# Neoadjuvant palbociclib and endocrine therapy versus chemotherapy in ER + /HER2- breast cancer: a randomized phase II trial

Alexios Matikas [1,2,15] ✉, Evangelos Tzoras[1,15], Michail Sarafidis[1,15], Emmanouil G. Sifakis [1], Judith Björhle[2], Elin Barnekow[3,4], Sara Margolin[3,4], Erika Isaksson-Friman[5], Luisa Edman Kessler[5], Athanasios Zouzos [1], Hemming Johansson[1], Mats Hellström[2], Susanne Agartz[1], Per Grybäck [6], Dimitrios Salgkamis [1], Ioannis Zerdes [1], Kang Wang [1], Johan Hartman [1,7], Balazs Acs [1,7], Wenwen Sun[1,7], Ceren Boyaci[1,7], Guillermo Villacampa[8,9], Tomas Pascual [8,10,11,12], Joaquin Gavila[8,13], Aleix Prat [10,11,12], Charles Perou[14], Yvonne Brandberg[1], Jonas Bergh [1,2], Thomas Hatschek[1,2] & Theodoros Foukakis [1,2] ✉

In PREDIX LumB patients with estrogen receptor positive and human epidermal growth factor receptor negative (ER + /HER2-) breast cancer > 20 mm and/or with lymph node metastasis were randomized 1:1 to receive either paclitaxel weekly for 12 weeks followed by palbociclib and endocrine therapy for 12 weeks (arm A), or the reverse sequence (arm B). Primary endpoint is objective radiologic response at 12 weeks ($ORR_{12}$), and key secondary endpoints are $ORR_{24}$, pathologic complete response, event-free survival, safety and correlative studies of tissue and circulating biomarkers. Whole exome sequencing and RNA sequencing were performed on baseline fresh frozen tissue samples. In total, 179 patients comprise the intention-to-treat population. There is no statistically significant difference between the two arms in $ORR_{12}$ (59% vs 45%, p = 0.058). An exploratory gene expression analysis identified differentially expressed genes and gene sets between responders and non-responders at 12 weeks. A predictive signature, CDKPredX, comprising 31 genes related to proliferation, ER signaling and immune activity was developed to identify patients resistant to chemotherapy but responding to palbociclib plus endocrine therapy ($p_{interaction}$=0.03). The predictive signature was independently validated in the CORALLEEN trial ($p_{interaction}$=0.048). Clinical-trials.gov identifier: NCT02603679

Inhibitors of cyclin dependent kinases 4 and 6 (CDK4/6i) are approved at both first and second line for the treatment of advanced estrogen receptor positive and human epidermal growth factor receptor 2 negative breast cancer (ER + /HER2- BC), based on the results of multiple randomized trials that have shown statistically and clinically significant prolongation of progression free and overall survival[1]. Their efficacy at the metastatic setting has inevitably led to their evaluation and subsequent approval in the adjuvant setting, where the addition to endocrine therapy of abemaciclib for two years or ribociclib for three years postoperatively has been shown to improve invasive disease-free

and distant disease-free survival[2,3]. The uptake of CDK4/6i as preoperative treatment has been slower. CDK4/6i in combination with endocrine therapy has been compared with either endocrine therapy alone or with neoadjuvant chemotherapy[4]. Efficacy measured with traditional endpoints such as pathologic complete response (pCR) has not been encouraging and focus has shifted to biomarkers of response, such as complete cell cycle arrest (CCCA)[5] or gene expression profiling-based risk classification[6,7]. Another clinically meaningful endpoint is tumor shrinkage to enable breast conserving surgery[8], since similar objective radiologic response (ORR) rates to CDK4/6i plus endocrine therapy and chemotherapy are seen among patients with ER + /HER2- BC[7], albeit with considerably less toxicity. However, the lack of predictive biomarkers undermines the implementation of neoadjuvant CDK4/6i[6].

An overlooked aspect of CDK4/6i integration to the preoperative setting is its timing in relation to cytotoxic chemotherapy. The addition of the CDK4/6i trilaciclib to chemotherapy confers a protective effect on bone marrow progenitors, which has been shown in randomized trials to reduce myelotoxicity during treatment for small cell lung cancer[9], and improve survival of patients with metastatic triple negative BC[10]. In the latter trial, trilaciclib was administered either simultaneously (arm B) or one day before chemotherapy (arm C) and in both cases efficacy improved over chemotherapy alone, an effect thought to be mediated by promotion of antitumor immunity. This hypothesis is in contrast with evidence on pancreatic cancer, where palbociclib after taxane chemotherapy augmented the cytotoxic effect by repressing DNA repair mechanisms, whereas palbociclib prior to chemotherapy mitigated its effect due to inhibition of tumor cell proliferation[11]. Considering the above, the identification of potential responders to preoperative CDK4/6i and the sequence of CDK4/6i and chemotherapy during preoperative treatment for ER + /HER2- BC may be of clinical importance[12]. The objectives of the randomized PREDIX LumB trial were to compare neoadjuvant paclitaxel with palbociclib and endocrine therapy in terms of objective response to treatment and to investigate their optimal sequence.

Here, we show the primary efficacy, safety and biomarker analyses of PREDIX LumB. There is no difference in terms of radiologic response between paclitaxel and palbociclib plus endocrine therapy, while the sequence of the treatments does not affect patient survival. In addition, we describe the development and validation of a transcriptomic biomarker that identifies breast tumors resistant to chemotherapy that retain sensitivity to palbociclib plus endocrine therapy.

## Results

### Patient characteristics

Between March 18th 2016 and July 29th 2021, 181 patients were enrolled in the trial. Two patients are excluded from all analyses, one because no protocol-mandated research core biopsy was obtained prior to treatment start and one because the results of the baseline computed tomography (CT) scans became available after treatment started and they showed disseminated disease. As such, the intention-to-treat (ITT) population comprises 179 patients. The available biopsy samples for translational analysis are summarized in Fig. 1.

The patients' baseline characteristics are shown in Table 1. Baseline characteristics were mostly balanced, except for median Ki67 (arm A 30% versus arm B 25%). -two thirds of all patients had histologically or cytologically verified lymph node metastasis prior to treatment start and more than 80% had progesterone receptor positive tumors.

### Primary and secondary efficacy endpoints

In total, 51 patients treated with paclitaxel and 40 treated with palbociclib and endocrine therapy as first treatment attained an objective response. The difference between the two groups in terms of the primary endpoint of radiological response at 12 weeks (ORR$_{12}$) was not statistically significant, 59% versus 45% ($p = 0.058$). There was no statistically significant difference in radiological response at 24 weeks (ORR$_{24}$) between the two arms (arm A 78% versus arm B 71%, $p = 0.293$). Eight patients attained pCR, three in arm A and five in arm B ($p = 0.469$). Residual cancer burden (RCB) class could be calculated for 132 patients, with most missing data due to prechemotherapy removal of a positive sentinel node, which was routine practice during the earlier years of the trial. RCB 0/I was observed in 11 patients treated in arm A ($n = 3$ RCB 0, $n = 8$ RCB 1) versus 5 in arm B ($n = 5$ RCB 0, $p = 0.073$). The overall distribution of RCB classes differed between the two arms ($p = 0.023$), with 59% in arm A versus 65% in arm B assessed as RCB-II and 24% versus 28% as RCB-III.

At the time of data cut-off (April 15th 2024), median follow-up time was 4.52 years (interquartile range, 3.39 – 5.59 years). There were 39 first events: 6 locoregional recurrences, 16 distant recurrences, 1 contralateral breast cancer, 7 disease progressions during neoadjuvant therapy, 7 second (new, non-breast) cancers and 2 deaths. There was no difference in terms of EFS between the two arms (HR = 1.06, 95% CI 0.57–1.99, $p = 0.86$). The corresponding five-year event-free survival (EFS) rates were 80.6% for arm A and 79.7% for arm B (Supplementary Fig. 1A). In addition, there were 24 post-surgery relapses, of which 7 were locoregional, 16 distant and 1 contralateral breast cancer. Relapse-free survival (RFS) did not differ between the two arms (HR = 0.84, 95% CI 0.37–1.87, $p = 0.66$; five-year rates 81.2% versus 86.0%; Supplementary Fig. 1B). Finally, there were 11 deaths, of which 6 breast cancer-related and 5 due to unrelated causes of death. No difference in overall survival (OS) between the two groups was observed (HR = 1.22, 95% CI 0.37–4.01, $p = 0.74$; five-year rates: 95.7% versus 93.2%; Supplementary Fig. 1C).

### Histologic, imaging and molecular biomarkers of response to treatment

Baseline pathologist-read and digital Ki67, pathologist-read tumor infiltrating lymphocytes (TILs) and positron emission tomography/computed tomography (PET/CT) calculated maximal Standardized Uptake Values (SUVmax) were associated with likelihood of objective response at 12 weeks regardless of treatment arm ($p_{interaction} > 0.1$ in all cases; Fig. 2A), although interaction between digital TILs and treatment arm was observed ($p = 0.037$).

CCCA was not statistically significantly different between the two arms at surgery ($p = 0.080$) or at 12 weeks ($p = 0.076$). Higher digital Ki67 at 12 weeks was associated with lower likelihood of pCR only for patients treated with paclitaxel (OR = 0.94, 95% CI 0.88–1.00, $p = 0.047$), but not with palbociclib (OR = 1.02, 95% CI 0.96–1.08, $p = 0.566$; $p_{interaction} = 0.063$).

At baseline there was no difference in median SUVmax according to PET/CT between the two treatment groups (5.0 versus 6.4, $p = 0.925$). Metabolic activity was better suppressed during treatment with paclitaxel than with palbociclib (SUVmax week 12/baseline ratio 0.40 versus 0.53, corresponding to 60% versus 47% relative decrease, $p = 0.009$). SUVmax week12/baseline ratio was associated with likelihood of objective response at 24 weeks (OR = 0.11, 95% CI 0.02–0.60, $p = 0.01$) and pCR (OR = 0.02, 95% CI 0.00–0.95, $p = 0.047$). In all cases, no differential effect according to treatment was noted ($p_{interaction} > 0.1$).

The associations between likelihood of objective response at 12 weeks and intrinsic subtypes, risk of recurrence-subtype-proliferation (ROR-P) score groups, *PI3K/AKT/mTOR* pathway mutations and the integrative clusters are summarized in Fig. 2B. No interaction with treatment was observed for intrinsic subtype ($p_{interaction} = 0.064$), ROR-P score groups ($p_{interaction} = 0.289$), *PI3K/AKT/mTOR* pathway alterations ($p_{interaction} = 0.085$), and integrative clusters ($p_{interaction} = 0.352$). The same was true for mutations or copy number alterations of key genes involved in *CDK4/6-Rb* pathway (Supplementary Fig. 2).

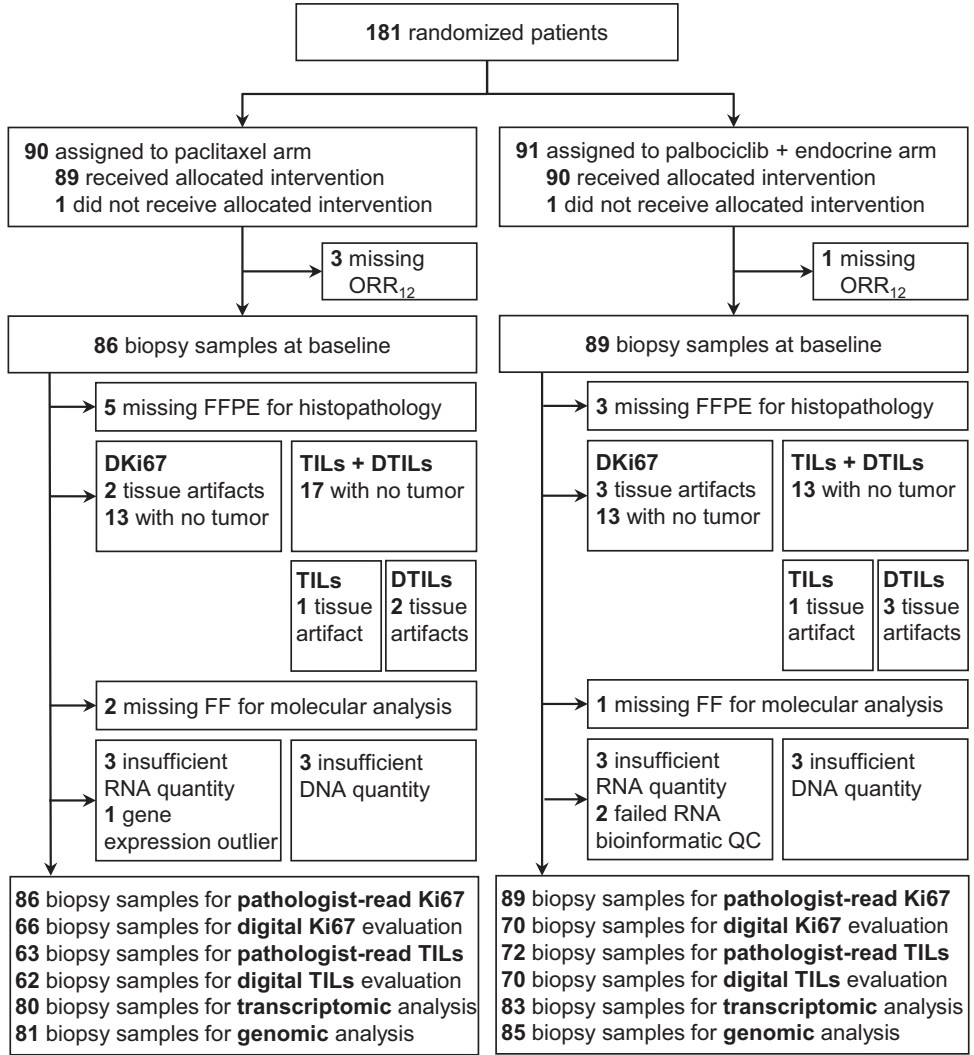

**Fig. 1 | CONSORT diagram of the PREDIX LumB trial.** Patient flow diagram, data availability per modality and reasons for missing data.

## Transcriptomic determinants of treatment response

Differential gene expression and gene set enrichment analyses of baseline RNA sequencing data revealed distinct molecular profiles associated with treatment response. Tumors from responders to paclitaxel exhibited upregulation of immune-related pathways and downregulation of estrogen receptor (ER)-related pathways. In contrast, responders to palbociclib and endocrine therapy showed enrichment of proliferation and cell cycle-related pathways (Supplementary Fig. 3). These findings were corroborated by analyses of relevant genes and molecular signatures. Lower expression levels of established ER-related gene signatures were associated with response to paclitaxel (Supplementary Fig. 4). A similar trend was observed in tumors with low proliferative activity, as indicated by low ROR-P scores and classification as the luminal A intrinsic subtype (Fig. 2).

Regarding the immune microenvironment, paclitaxel responders demonstrated expression patterns indicative of an anti-tumor immune response, including higher levels of antigen presentation (Supplementary Fig. 5), increased cytotoxic or effector T cells (Supplementary Fig. 6), and higher levels of humoral immunity (Supplementary Fig. 7). Similar associations were observed when assessing the immune-related modules derived in silico from the publicly available gene sets of the Breast Cancer 360™ (BC360) panel (Supplementary Fig. 8). As expected due to the study design involving treatment reassignment after 12 weeks, patients achieving RCB 0/1 had baseline tumors

characterized by upregulated proliferation and immune-related pathways and downregulated ER-related gene sets (Supplementary Fig. 9). Importantly, the recurrence of these biological patterns in responders, regardless of the time point of assessment (week 12 or surgery), underscores the time-independent nature of treatment response biology.

## Development and validation of CDKPredX predictor

To quantify the relationship between these biological phenotypes and treatment response, a metagene predictor termed CDKPredX was developed. CDKPredX+ subgroup was defined by tumors with high levels of proliferation and ER pathway expression, and low levels of immune-related gene expression, and comprised 42 of the 163 patients (25.8%). CDKPredX+ was enriched for luminal B intrinsic subtype ($p < 0.001$), exhibited high expression of the retinoblastoma loss-of-function signature (RBsig) ($p < 0.001$) and was associated with non-lobular histology ($p = 0.026$). These findings were further validated in the SCAN-B cohort (Supplementary Fig. 10). In our cohort, expression of RBsig was strongly correlated with E2F signature expression ($R = 0.97$, $p < 0.001$). There were no *RB1* point mutations, while deep deletions of *RB1* gene were a rare event (5/166 patients, 3%) (Supplementary Fig. 11). The clinicopathologic and molecular characteristics of the CDKPredX subgroups are summarized in Fig. 3A and Supplementary Table 1. Further exploratory analysis compared unscaled continuous 21-gene recurrence score distributions, computed using a

**Table 1 | Baseline patient characteristics**

|  | Arm A N = 89 (100%) | Arm B N = 90 (100%) |
|---|---|---|
| **Median Age (interquartile range)** | 54 (47 – 61) | 56 (47 – 66) |
| **Menopausal status** | | |
| Premenopausal | 35 (38.8%) | 35 (38.8%) |
| Postmenopausa | 51 (57.3%) | 51 (56.6%) |
| Unknown | 3 (3.3%) | 4 (4.4%) |
| **Histologic type** | | |
| Ductal | 63 (70.7%) | 59 (65.5%) |
| Lobular | 18 (20.2%) | 22 (24.4%) |
| Other or unknown | 8 (9.0%) | 9 (10.0%) |
| **Grade** | | |
| I and II | 68 (76.4%) | 77 (85.5%) |
| III | 21 (23.6%) | 12 (13.3%) |
| Unknown | 0 | 1 (1.1%) |
| **Tumor size (mm)** | | |
| ≤ 20 | 15 (16.8%) | 7 (18.8%) |
| 21 – 50 | 57 (64.0%) | 51 (56.6%) |
| >50 | 17 (19.1%) | 122 (24.4%) |
| **Nodal status (cN)** | | |
| Negative | 32 (35.9%) | 28 (31.1%) |
| Positive | 57 (64.0%) | 62 (68.8%) |
| **Progesterone receptor** | | |
| Negative | 15 (16.8%) | 11 (12.2%) |
| Positive | 74 (83.1%) | 79 (87.7%) |
| **Median Ki67 % (interquartile range)** | 30 (21–40) | 25 (20–33) |
| **Intrinsic Subtypes[1]** | | |
| Luminal A | 49 (61.2%) | 49 (59.0%) |
| Luminal B | 27 (33.8%) | 32 (38.6%) |
| HER2-Enriched or Basal-like | 4 (5%) | 2 (2.4%) |
| **PI3K/AKT/mTOR mutations[2]** | | |
| No | 48 (59.3%) | 44 (51.8%) |
| Yes | 33 (40.7%) | 41 (48.2%) |
| **Type of endocrine therapy[3]** | | |
| Tamoxifen | 31 (34.8%) | 34 (37.7%) |
| Aromatase inhibitor | 59 (66.3%) | 58 (65.5%) |
| Goserelin | 16 (18.0%) | 11 (12.2%) |
| **Type of surgery** | | |
| Breast conserving | 54 (60.7%) | 52 (57.8%) |
| Mastectomy | 32 (35.9%) | 35 (38.9%) |
| Unknown | 3 (3.4%) | 3 (3.3%) |
| **Adjuvant epirubicin and cyclopho- sphamide administered** | | |
| Completed three cycles | 82 (92.1%) | 82 (91.1%) |
| Completed fewer than three cycles | 3 (3.4%) | 2 (2.2%) |
| Did not receive any cycles | 3 (3.4%) | 4 (4.4%) |
| Unknown | 1 (1.1%) | 2 (2.2%) |

[1] N = 163
[2] N = 166
[3] The same patient may have received more than one endocrine therapies.

research-based implementation, between CDKPredX-defined subgroups (Supplementary Fig. 12).

A statistically significant interaction between CDKPredX subgroups and treatment was observed ($p_{interaction}$=0.03; Fig. 3B), as the predictor selected responders to palbociclib plus endocrine treatment and resistant to chemotherapy. $ORR_{12}$ for CDKPredX+ tumors was 47.8% for CDK4/6i and endocrine treatment versus 31.6% for chemotherapy (OR = 1.99, 95% CI 0.57–7.38, $p = 0.288$), and 43.3% versus 65.6%, respectively for CDKPredX- tumors (OR = 0.4, 95% CI 0.19–0.83, $p = 0.015$). This interaction was confirmed in the COR-ALLEEN trial ($p_{interaction}$= 0.048; Fig. 3B), with the corresponding OR for the CDKPredX+ and CDKPredX- groups being OR = 3, 95% CI 0.28 – 70.88 ($p = 0.395$) in favor of palbociclib and endocrine therapy, and OR = 0.19, 95% CI 0.04 – 0.71 ($p = 0.019$) in favor of chemotherapy. This effect was noted regardless of menopausal status (Supplementary Fig. 13). CDKPredX+ was not statistically significantly associated with $ORR_{24}$, RCB, pCR rates and time-to-failure endpoints in PREDIX LumB (Supplementary Figs. 14 and 15). The association of CDKPredX and PET/CT SUVmax with outcomes in the subgroup of interest of lobular tumors is shown in Supplementary Fig. 16.

The chemoresistance of CDKPredX+ patients was further confirmed in the I-SPY2 and SCAN-B cohorts. CDKPredX+ showed lower pCR rates to neoadjuvant chemotherapy with or without immunotherapy compared to CDKPredX- in I-SPY2 (OR = 0.15, 95% CI 0.04–0.43, $p = 0.002$) (Supplementary Fig. 17) and derived no benefit from adjuvant chemotherapy in SCAN-B trial (inverse probability treatment weighting-adjusted $p = 0.61$). CDKPredX was not prognostic for long-term outcomes regardless of treatment (Supplementary Fig. 18).

## Safety and health related quality of life
There were no differences between the two treatment groups in terms of any dose reduction (43.8% versus 50.0%, $p = 0.41$), any dose delay (59.6% versus 51.1%, $p = 0.26$) or treatment discontinuation of at least one part of the preoperative treatment (27.0% versus 34.4%, $p = 0.28$) for arm A versus arm B, respectively. Specifically, in arm A 19.1% and 9.0% prematurely discontinued paclitaxel and palbociclib respectively, while the corresponding proportions of patients in arm B were 28.8% for paclitaxel and 5.5% for palbociclib. The most common adverse events of any grade during paclitaxel were peripheral sensory neuropathy, rash and fatigue and during palbociclib neutropenia and nausea. The most common grade 3/4 adverse events during paclitaxel were peripheral sensory neuropathy and neutropenia, and during palbociclib neutropenia (Table 2). There were no differences in adverse events of any grade or of grade 3/4 between the two treatment arms.

The summary of the patient reported outcomes is shown in Fig. 4. The response rate of the health-related quality of life (HRQoL) questionnaires was 98% at baseline and ranged from 82-94% during treatment up to one year postoperatively (18 months since enrollment to the trial). At baseline, there was no difference between the two treatment groups (Supplementary Fig. 19). HRQoL was better preserved under treatment with palbociclib than paclitaxel during the first 12 weeks, with a shift after treatment switch in several domains such as dyspnea, diarrhea, and hair loss (Fig. 4 and Supplementary Figs. 20 and 21). At the time of the first annual follow-up visit, patient-reported HRQoL had returned to baseline levels for most variables regardless of randomization arm. However, compared with Swedish general population normative data, moderate differences in cognitive function, social function and insomnia and small differences in role function, fatigue and financial problems were observed (Supplementary Fig. 22).

## Discussion
Consistent with previously reported results from randomized trials, in PREDIX LumB we did not observe any difference between chemotherapy and CDK4/6i plus endocrine therapy for the primary endpoint of $ORR_{12}$. Toxicity was in line with prior studies for both palbociclib/endocrine therapy and paclitaxel, with no new safety concerns. HRQoL was better preserved during treatment with palbociclib, regardless of treatment sequence, and HRQoL levels returned to baseline at one year postoperatively in most, though not all, domains.

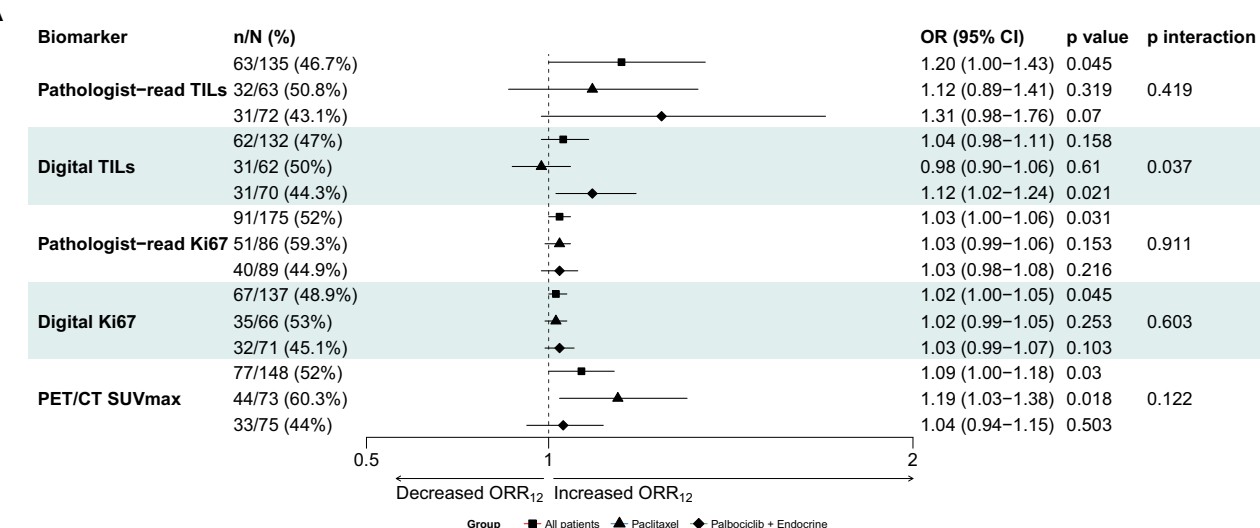

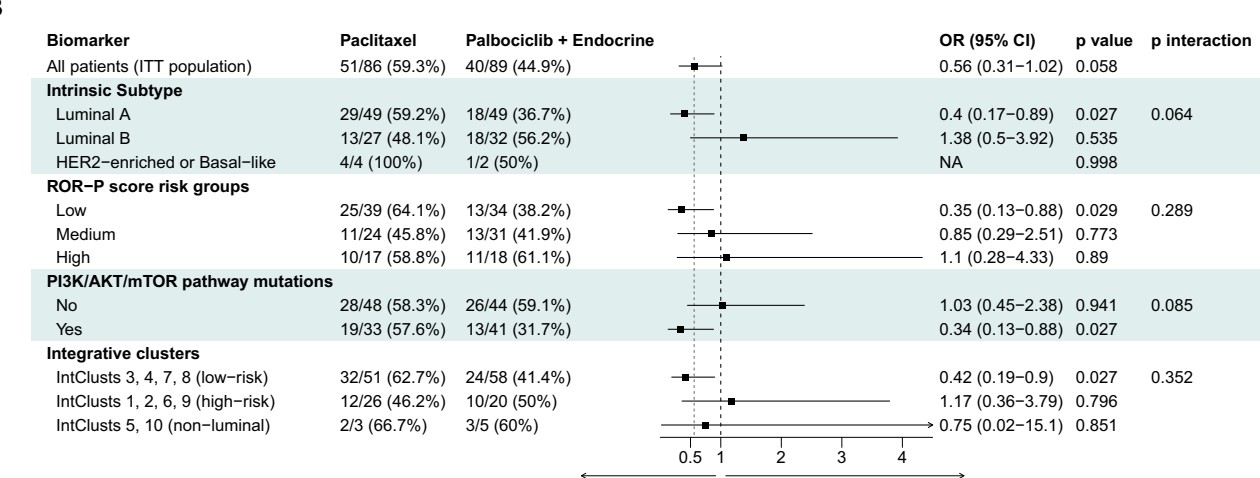

**Fig. 2 | Histopathologic and molecular predictors of ORR₁₂ in PREDIX LumB.** Forest plot for the primary endpoint of $ORR_{12}$ according to histopathologic biomarkers and PET/CT SUVmax as continuous variables (**A**), and molecular biomarkers subgroup analysis (**B**) in PREDIX LumB. Odds ratios were estimated using logistic regression with two-sided Wald tests, interaction effects were evaluated using two-sided likelihood ratio tests, and no adjustment for multiple comparisons was applied. Source data are provided as a Source Data file.

While prior evidence suggested enhanced cytotoxic effect when CDK4/6i followed chemotherapy compared to the reverse sequence, this was not confirmed in PREDIX LumB. Considering these findings, alternative strategies for integrating CDK4/6i into the preoperative management of ER + /HER2- BC should be pursued, particularly through the identification of predictive biomarkers to improve patient selection, which was the trial's key secondary objective.

Besides its clinically oriented aims, PREDIX LumB has a strong translational focus with integration of metabolic imaging, longitudinal tissue and liquid biopsy sampling during treatment. In this study, baseline histology and molecular biomarkers were generally prognostic for radiologic response to treatment, but not predictive of differential treatment effect. Regarding metabolic response to treatment, small retrospective studies have previously evaluated the role of PET/CT in assessing response to CDK4/6i for metastatic breast cancer[13,14]. Here, we integrated serial PET/CT scans during neoadjuvant CDK4/6i within the context of a randomized trial and demonstrated the prognostic significance of SUVmax suppression regardless of administered treatment. The clinical utility of our exploratory findings in terms of early assessment of response and treatment adaptation should be further evaluated prospectively.

Building on our previous findings linking immune-related gene expression to chemosensitivity in ER + /HER2- BC[15,16], the fact that commercial gene signatures consist of ER-signaling and proliferation modules[17], the added prognostic value of immune gene expression to these signatures[18], and the transcriptomic findings in PREDIX LumB, we developed the CDKPredX predictor by interrogating the transcriptome of untreated tissue samples. Across multiple independent cohorts, including two randomized trials directly comparing neoadjuvant CDK4/6i with chemotherapy, CDKPredX identified a consistent subgroup of patients with clinically (as in PREDIX LumB) or molecularly (as in CORALLEEN) high-risk, ER + /HER2- BC enriched for responders to CDK4/6i and endocrine therapy. This subgroup exhibited low rates of radiologic and pathologic response to neoadjuvant chemotherapy, derived little benefit from adjuvant chemotherapy, and lacked sensitivity to immune checkpoint inhibitors, which are emerging as a new treatment option for a subgroup of patients with ER + / HER2- BC[19,20]. CDKPredX is a transcriptomic signature that predicts resistance to chemotherapy while the effect of CDK4/6i and endocrine therapy is retained or even enriched, following several studies that had failed to demonstrate the predictive value of clinicopathologic[21] and multiple genomic and transcriptomic biomarkers[6,22–24]. The lack of

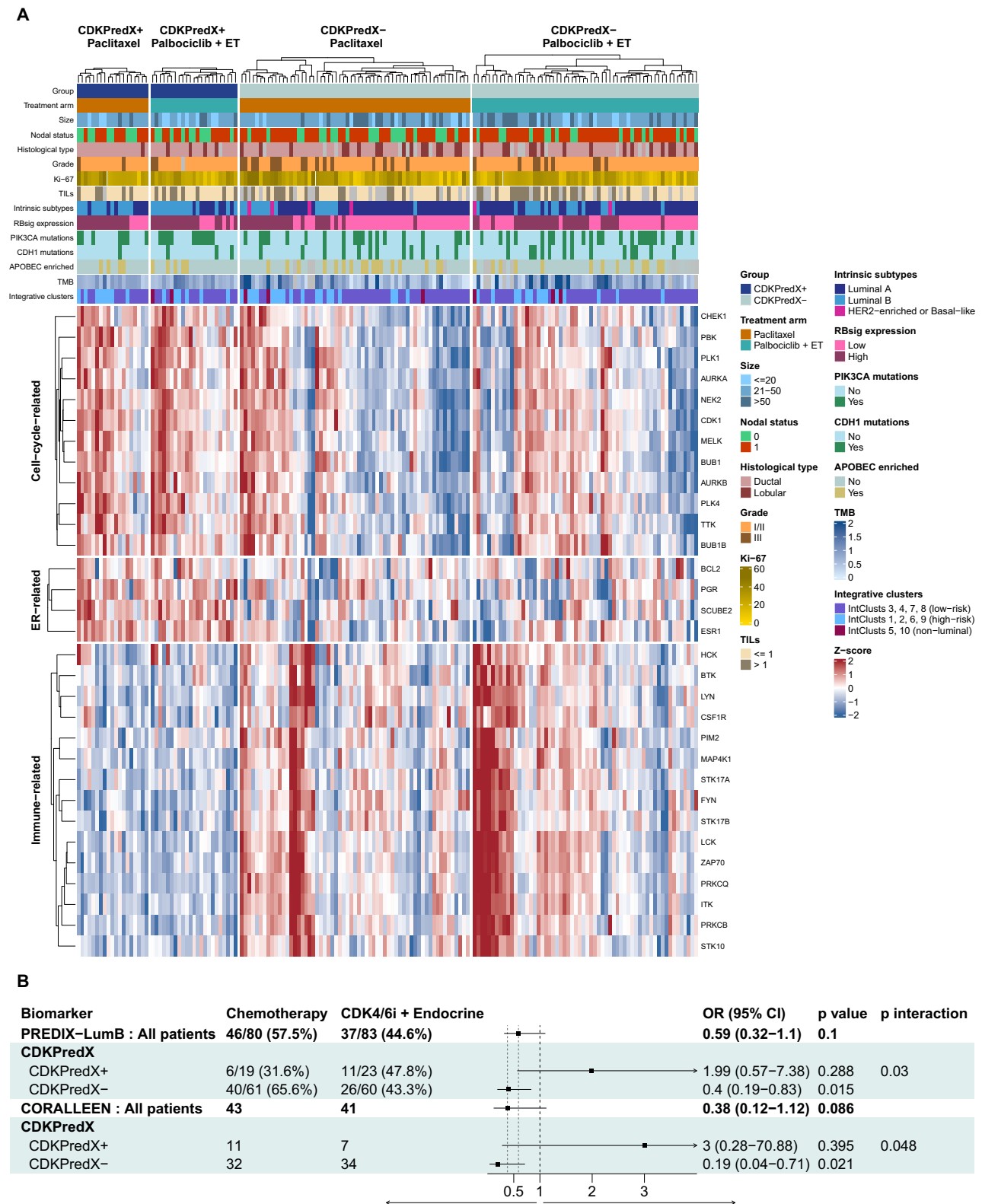

**Fig. 3 | CDKPredX stratification and association with ORR₁₂.** Clinical, histopathological and molecular characteristics of the CDKPredX+ and CDKPredX- groups in PREDIX LumB (**A**). Forest plots for the primary endpoint of ORR₁₂ according to the CDKPredX predictor in PREDIX LumB and CORALLEEN (**B**). Odds ratios were estimated using logistic regression with two-sided Wald tests, interaction effects were evaluated using two-sided likelihood ratio tests, and no adjustment for multiple comparisons was applied. Source data are provided as a Source Data file.

**Table 2 | a Patients with any predefined adverse event of any grade by allocated treatment arm and drug (intention-to-treat population) and comparison between the two arms. b Patients with any predefined adverse event grade 3 or 4 by allocated treatment arm and drug (intention-to-treat population) and comparison between the two arms**

| a | | | | | |
|---|---|---|---|---|---|
| | Arm A | | Arm B | | P value |
| | Paclitaxel N = 89 | Palbociclib N = 89 | Paclitaxel N = 90 | Palbociclib N = 90 | |
| Allergic Reaction | 3 (3.3%) | 0 | 0 | 1 (1.1%) | 0.249 |
| Alopecia | 1 (1.1%) | 0 | 0 | 2 (2.2%) | 0.626 |
| Anemia | 0 | 0 | 1 (1.1%) | 0 | 0.846 |
| Constipation | 0 | 0 | 0 | 0 | 1.000 |
| Diarrhea | 2 (2.2%) | 0 | 0 | 3 (3.3%) | 0.496 |
| Elevated liver transaminases | 0 | 2 (2.2%) | 2 (2.2%) | 0 | 0.308 |
| Fatigue | 10 (11.2%) | 0 | 0 | 6 (6.6%) | 0.456 |
| Headache | 1 (1.1%) | 1 (1.1%) | 1 (1.1%) | 0 | 0.405 |
| Oral mucositis | 3 (3.3%) | 2 (2.2%) | 1 (1.1%) | 4 (4.4%) | 0.180 |
| Myalgia | 0 | 0 | 0 | 1 (1.1%) | 1.000 |
| Nausea | 0 | 2 (2.2%) | 5 (5.5%) | 3 (3.3%) | 0.846 |
| Neutropenia | 2 (2.2%) | 44 (49.4%) | 51 (56.6%) | 1 (1.1%) | 0.873 |
| Peripheral sensory neuropathy | 44 (49.4%) | 1 (1.1%) | 0 | 47 (52.2%) | 0.433 |
| Rash | 14 (15.7%) | 3 (3.3%) | 1 (1.1%) | 9 (9.9%) | 0.236 |
| Leukopenia | 5 (5.6%) | 7 (7.8%) | 8 (8.8%) | 2 (2.2%) | 1.000 |
| b | | | | | |
| | Arm A | | Arm B | | P value |
| | Paclitaxel N = 89 | Palbociclib N = 89 | Paclitaxel N = 90 | Palbociclib N = 90 | |
| Allergic Reaction | 1 (1.1%) | 0 | 0 | 1 (1.1%) | 1.000 |
| Anemia | 1 (1.1%) | 0 | 0 | 0 | 0.497 |
| Diarrhea | 1 (1.1%) | 0 | 3 | 0 | 1.000 |
| Elevated liver transaminases | 0 | 0 | 0 | 1 (1.1%) | 0.121 |
| Fatigue | 1 (1.1%) | 0 | 0 | 1 (1.1%) | 1.000 |
| Headache | 1 (1.1%) | 0 | 0 | 0 | 0.497 |
| Nausea | 0 | 0 | 0 | 1 (1.1%) | 1.000 |
| Neutropenia | 5 (5.6%) | 32 (35.9%) | 35 (38.8%) | 3 (3.3%) | 1.000 |
| Peripheral sensory neuropathy | 4 (4.5%) | 0 | 0 | 4 (4.4%) | 1.000 |
| Rash | 1 (1.1%) | 0 | 0 | 0 | 0.497 |
| Leukopenia | 1 (1.1%) | 1 (1.1%) | 1 (1.1%) | 0 | 0.621 |

response to the current and emerging standard of care at the neoadjuvant setting, even among molecularly selected high-risk patients[25], marks a population with a clearly unmet clinical need and paves the way for the rational, evidence-based implementation of preoperative CDK4/6i. Further studies will shed light on the clinical validity and utility of CDKPredX in the adjuvant and metastatic settings where CDK4/6i have an established role, albeit concerning all-comers without biomarker-driven patient selection. This lack of precision may lead to both unnecessary toxicity and high costs for patients unlikely to respond. Beyond the clinical implications of this study, our results also highlight the biologic heterogeneity of ER + /HER2- BC and confirm prior findings on therapy-driven subtyping by leveraging multi-omics analyses[26–28], marking a paradigm shift away from the currently recommended one-size-fits-all approach[29].

The relatively small sample size of this randomized trial was in line with prior trials studying neoadjuvant CDK4/6i[5,7,30,31], however it may have been insufficient to detect small differences between administered treatments or associations of biomarkers with outcomes, whereas the trial was not powered for comparing the two treatment sequences for time-to-failure outcomes. Other limitations that should be acknowledged are the short treatment duration of 12 weeks with

palbociclib and endocrine therapy which may have been insufficient to exert its full antitumoral effect and the relatively short follow-up considering the long natural history of ER + /HER2- BC, although it's one of the longest reported among neoadjuvant CDK4/6i trials. Furthermore, due to the design of PREDIX LumB it is not possible to discern the relative contributions of palbociclib or endocrine therapy to treatment effect and biomarker associations. In addition, the primary endpoint of radiologic response before treatment switch was a pragmatic choice considering the low pCR rates in PREDIX LumB and other trials[7,30] and the expected small number of time-to-failure events due to the sample size. However, radiologic response is a clinically relevant endpoint, since tumor shrinkage facilitates surgical excision and de-escalation of breast surgery[8]. Finally, although we took appropriate steps to externally validate our findings, we acknowledge that differences between the study cohorts in terms of chemotherapy regimens, administered CDK4/6 inhibitor in CORALLEEN, study endpoints and methods of assessment of response may have influenced our results.

In conclusion, using tissue samples obtained from patients treated within the academic randomized phase 2 PREDIX LumB trial, we describe the development and external validation of a transcriptomic

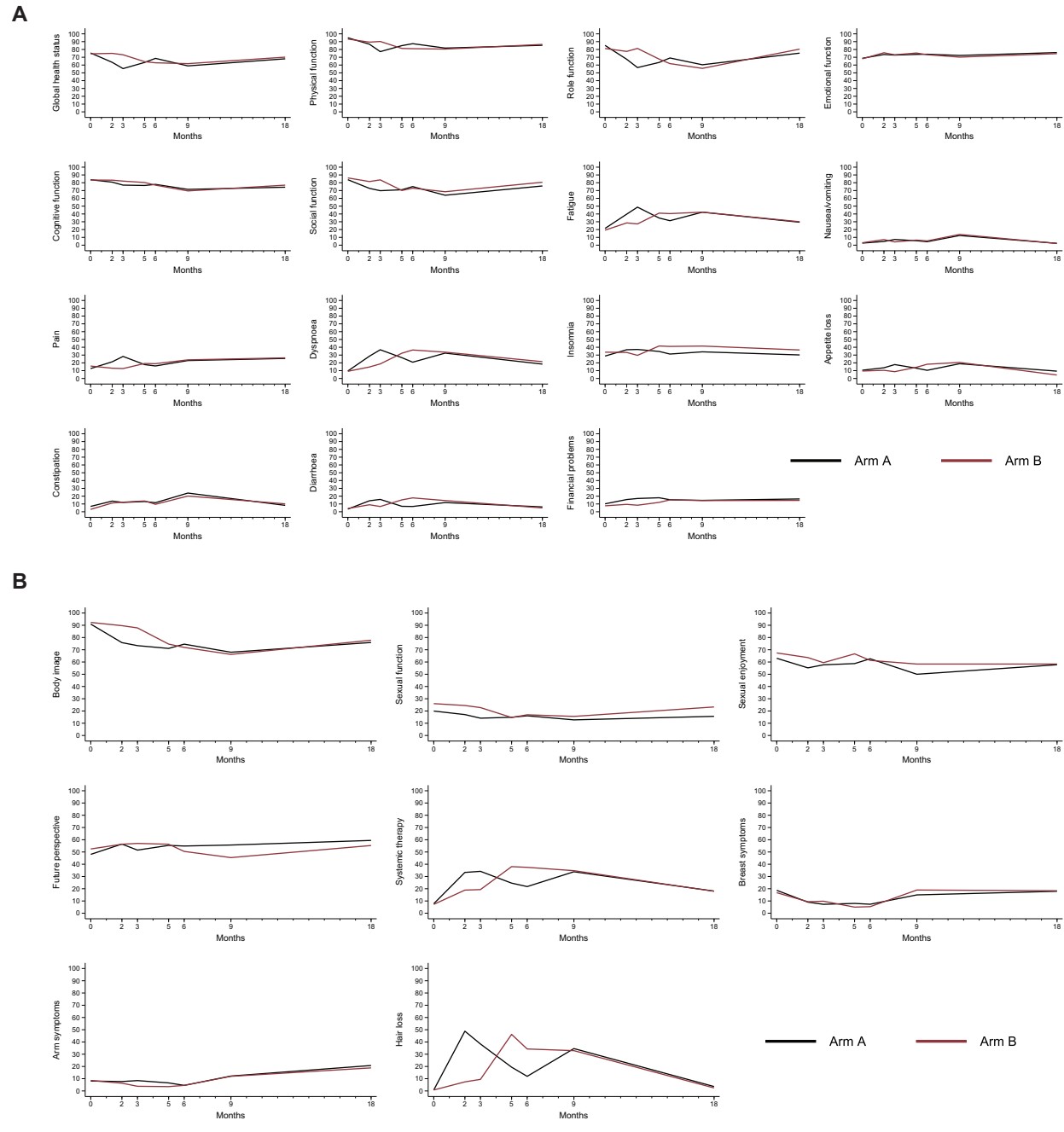

**Fig. 4 | Longitudinal health-related quality of life by treatment arm.** Longitudinal comparison of health-related quality of life data between the two treatment arms, according to the EORTC QLQ-C30 (**A**) and the EORTC BR23 (**B**) questionnaires.

signature, CDKPredX, which predicts radiologic response to CDK4/6i and endocrine therapy, and resistance to chemotherapy for patients with ER + /HER2- BC at the neoadjuvant setting. The development of CDKPredX as a single sample predictor is ongoing and a planned prospective trial will assess its clinical utility.

## Methods

### Study oversight
This study was approved by the Regional Ethical Committee in Stockholm (dnr 2014/1492-31/4) and the Swedish Medical Product Agency. All patients provided written informed consent before

inclusion. The study was conducted according to the Declaration of Helsinki and the principles of good clinical practice and was registered with EudraCT number 2014-000810-72 and at the ClinicalTrials.gov website, identifier NCT02603679.

### Study design and participants
PREDIX LumB is an academic, prospective, randomized, open-label, multicenter phase 2 trial which was conducted at three centers in Stockholm, Sweden (Karolinska University Hospital, Södersjukhuset and Capio S:t Göran Hospital). The trial protocol is available as supplementary material. The clinical trial is reported hereunder according

to the Consolidated Standards of Reporting Trials (CONSORT) guidelines[32] and the correlative analyses according to the Reporting Recommendations for Tumor Marker Prognostic Studies (REMARK) guidelines[33].

Eligible patients were women and men over 35 years with Eastern Cooperative Oncology Group performance status 0 or 1, diagnosed with ER positive (ER ≥ 10%, in accordance with the Swedish National Care Program for breast cancer) and HER2-negative BC, greater than 2 cm in size and/or node-positive. Patients with at most two distant metastases that could be treated with curative intent could be enrolled. Adequate cardiac, renal and hepatic function, and no history of other malignancies during the past five years were required for inclusion to the study.

### Randomization and masking

Patients were randomly assigned (1:1) into the two treatment groups described hereunder. Randomization was conducted at the Central Trial Office at Karolinska University Hospital by a web-based procedure (TENALEA, TransEuropean Network for Clinical Trial Services, Amsterdam, Netherlands). Random assignment was stratified by participating site. Random permuted blocks of different sizes (block size of 2 or 4) were used to allocate the patients to each treatment group. This was an open-label study, thus both participants and investigators were aware of the treatment assignment.

### Procedures

Patients allocated to arm A received 12 cycles of paclitaxel 80 mg/m² administered weekly and then switched to treatment for 12 weeks with palbociclib (125 mg/day during days 1–21 every 28 days) in combination with endocrine therapy which was selected at the treating physician's discretion (anastrozole 1 mg/day or letrozole 2.5 mg/day or exemestane 25 mg/day or tamoxifen 20 mg/day). Premenopausal patients planned for treatment with an aromatase inhibitor received additional goserelin 3.6 mg every four weeks subcutaneously. Patients allocated to arm B were treated with the reverse sequence, first 12 weeks of palbociclib and endocrine therapy, followed by 12 cycles of weekly paclitaxel. In case of disease progression before the 12-week timepoint, patients could prematurely switch to the other treatment. Breast surgery was performed ~two weeks after the final cycle. All patients received three adjuvant cycles of epirubicin 75–100 mg/m² and cyclophosphamide 600 mg/m² every three weeks, followed by radiotherapy and endocrine therapy in accordance with national guidelines and local practice.

Patients were followed during treatment with office visits for clinical examination and routine laboratory assessments (complete blood count, serum chemistry) every four weeks during the neoadjuvant phase. Response to treatment was assessed with breast imaging at 12 (prior to treatment switch) and 24 weeks (before surgery), either with mammography and ultrasound or with magnetic resonance imaging (MRI) of the breasts. Fluorine 18–labeled fluorodeoxyglucose (18F-FDG) combined positron emission tomography/computed tomography (PET/CT) was performed at baseline within two weeks prior to treatment start and at the 12-week timepoint prior to treatment switch. Following intravenous FDG injection and a 60-minute uptake phase, a combined PET/CT scan was obtained from the thorax and regional lymph nodes to limit radiation exposure. Within the scope of this analysis, the evaluation of the combined PET/CT images was made by calculating the maximal Standardized Uptake Values (SUVmax).

Safety was assessed according to the National Cancer Institute Common Terminology Criteria for Adverse Events version 4.0. Health-related quality of life (HRQoL) was assessed using the European Organization for Research and Treatment of Cancer (EORTC) quality of life questionnaire C30 (QLQ-C30) and the breast cancer–specific BR23 questionnaire[34,35]. HRQoL forms were completed at baseline before

randomization, every six weeks during treatment (four timepoints), three months after surgery and annually at follow-up visits.

After the end of adjuvant chemotherapy and radiotherapy, patients were followed according to protocol for up to ten years with annual office visits for physical examination, documentation of adverse events and collection of HRQoL (the latter during years 0 – 5), complete blood count, serum chemistry, breast imaging, and plasma and serum collection for research use. At the time of recurrence further plasma and serum samples were collected, while a metastatic biopsy for research use was recommended but not obligatory.

### Outcomes

The primary endpoint of the study is the rates of locally assessed radiological response at 12 weeks (ORR₁₂), defined as complete (complete resolution of all lesions) or partial response (reduction in size of at least 30% with no new lesions). Secondary efficacy endpoints include ORR at 24 weeks (ORR₂₄); rates of locally assessed pCR, defined as absence of invasive carcinoma in the breast and axillary lymph nodes (ypT0/Tis, ypN0); pathologic response according to Residual Cancer Burden (RCB)[36], event-free survival (EFS), defined as time from randomization to disease progression, disease recurrence (local, regional, or distant), contralateral BC, other malignancy, or death from any cause, whichever occurs first; relapse-free survival (RFS), defined as time from surgery to disease recurrence (local, regional, or distant), or death from any cause, whichever occurs first; and overall survival (OS), defined as time from randomization to death from any cause. The definitions of the time-to-failure endpoints are generally aligned with the standardized NeoSTEEP (Standardized Definitions for Efficacy End Points) definitions[37]. Other secondary endpoints include safety and HRQoL as described above, including comparison with Swedish general population normative HRQoL data using EORTC QLQ-C30[38], and biomarker studies for markers of response and resistance to the study treatments.

### Study assessments

**Tissue collection.** Tissue biospecimens were collected from the patients prior to the initiation of the neoadjuvant therapy, as specified in the study protocol. Up to five 14-gauge ultrasound-guided core needle tissue biopsies were collected. Three fresh-frozen (FF) tissue samples were stored directly at −80 °C after collection. The remaining two samples were directly immersed into 10% neutral-buffered formalin for preparation of formalin-fixed paraffin-embedded (FFPE) tissue blocks. Biopsies at baseline and on-treatment were obtained after the PET/CT scan. Biopsies on-treatment were obtained at a median time of four days from the last administered palbociclib dose at week 12 (arm B), while surgery was performed at a median time of twelve days from the last administered palbociclib dose (arm A). Whole peripheral blood was also collected before the initiation of neoadjuvant therapy in 4 ml EDTA tubes and stored at −80 °C. Plasma and serum samples were collected at baseline and every six weeks during treatment, as well as during follow-up.

**Hematoxylin & eosin staining and Ki67 immunohistochemistry.** Whole tissue sections (4 µm thick) from FFPE research biopsies were stained by E.T. for hematoxylin and eosin (H&E) and Ki67 immunohistochemistry (IHC) (clone SP6, dilution 1:100). For immunohistochemistry, the IntelliPath FLX automated slide stainer was used (BioCare Medical, California, USA) according to manufacturer's protocol. Slides were deparaffinized with xylene for 20 min and rehydrated through a series of graded ethanol to water solutions. Antigen retrieval was performed with Diva Decloaker (BioCare Medical) in a decloaking chamber (BioCare Medical) under pressure at 95 °C for 15 min and cooled down in washing buffer. Peroxidase block, protein block, immunostaining, HRP-coupled detection, and DAB chromogen were carried out with MACH1 Universal HRP-Polymer Detection Kit

(BioCare Medical) and counterstaining with hematoxylin (Intellipath hematoxylin, BioCare Medical). After dehydration with increasing ethanol concentration, the slides were mounted using Pertex (Histolab, Gothenburg, Sweden).

**Manual evaluation of tumor infiltrating lymphocytes, immune aggregates and Ki67.** Stromal tumor infiltrating lymphocytes (TILs) were assessed by a certified breast pathologist (W.S.), who was blinded to other clinicopathological and molecular characteristics, as the percentage of tumor stroma covered by infiltrating lymphocytes, according to the recommendations of the International TILs Working Group[39]. The presence of immune aggregates was assessed by E.T. as a binary variable. The aggregates could be localized either inside or outside the invasive cancer area. A figure of representative immune aggregation morphology can be found in Supplementary Fig. 23. Expression of Ki67 (clone 30-9) at the baseline diagnostic biopsy and the surgical specimen was assessed locally according to routine care at the time the trial was conducted as the percentage of tumor cells with positive nuclear staining counted in a hot spot with a minimum of 200 tumor cells.

**Generation of digital images.** Digital images were generated by E.T. through slide scanning at 40x magnification with the NanoZoomer-XR digital slide scanner (cat # 12000, Hamamatsu Photonics, Hamamatsu, Japan) and the NDP.scan software (Hamamatsu Photonics, Hamamatsu, Japan) and were saved in NDPI file format.

**Digital image assessment.** Digital images were processed with the QuPath software[40]. The invasive cancer area was annotated in H&E images by W.S. according to TILs Working Group guidelines and in IHC by C.B. A cell classifier algorithm[41,42] was used to enumerate digital TILs (DTILs), and the variable easTILs%=TILs Cell Area/Stroma Area*100 was calculated as a surrogate of the respective definition from the TILs Working Group for the visual assessment. Across timepoints, pathologist-read and digital TILs were weakly but statistically significantly correlated (Spearman's rho=0.292, $p < 0.001$). The expression of digital Ki67 (DKi67) was automatically enumerated, as previously described[43–46], using the guideline from the International Ki67 in Breast Cancer Working Group (https://www.ki67inbreastcancerwg.org/). Regardless of the assessment method, complete cell cycle arrest (CCCA) was defined as Ki67 ≤ 2.7%[30]. Across timepoints, pathologist-read and digital Ki67 were moderately and statistically significantly correlated (Spearman's rho=0.553, $p < 0.001$).

**Nucleic acid extraction, library preparation, and sequencing.** Simultaneous isolation of bulk RNA and DNA from fresh-frozen tumor biopsies was performed by E.T. using the AllPrep DNA/RNA/Protein Mini Kit (cat # 80004, QIAGEN, Hilden, Germany) according to manufacturer's instructions. Each fresh-frozen sample was first minced into small pieces on a petri dish on dry ice using sterile stainless steel scalpels No.10 (cat # 0501, Swann-Morton, Sheffield, UK). The tissue pieces were disrupted and homogenized in a mixture of Buffer RLT with β-mercaptoethanol (cat # 125470100, Thermo Fisher Scientific, USA) using 2.8 mm and 1.4 mm zirconium oxide homogenization beads included in the Precellys® CKMix Tissue Homogenizing Kit (cat # P000918-LYSK0-A, Bertin Corp. Maryland, US), reinforced tubes (cat # D1031-RFS, Benchmark Scientific, Sayreville, US) and the BeadBug microtube homogenizer (cat # D1030E, Benchmark Scientific, Sayreville, US). After complete tissue disruption and during the RNA purification process, DNA digestion was performed using the RNase-free DNase set (cat # 79254, QIAGEN, Hilden, Germany). RNA was eluted with RNase-free water while DNA was eluted with EB buffer (cat # 19086, QIAGEN, Hilden, Germany). Germline DNA was isolated from whole peripheral blood using the FlexiGene DNA kit (cat # 51206, QIAGEN, Hilden, Germany). All nucleic acids were collected in Eppendorf® Forensic DNA Grade Safe-Lock 1.5 mL microcentrifuge tubes (Eppendorf SE, Hamburg, Germany). Initial quality control of the extracted nucleic acids was performed right after elution of the tumor RNA and DNA samples followed by storage at –80 °C. Quantification of RNA and DNA was performed by the Qubit 3.0 Fluorometer (cat # Q33216, Thermo Fisher Scientific, USA) using Qubit™ RNA BR Assay Kit (cat # Q10210) and Qubit™ dsDNA BR Assay Kit (cat # Q32850, Invitrogen, USA), respectively. Purity was assessed using the Nanodrop ND-1000 (Saveen Werner, Malmö, Sweden).

For library preparation, all nucleic acid samples were first prepared to a final concentration of 20 ng/μL, while the initial nucleic acid input was 100 ng. RNA libraries were prepared using the Illumina Stranded Total RNA library preparation kit with Ribo-Zero Plus treatment (cat # 20040525/20040529, Illumina Inc.). Unique dual indexes (cat # 20040553/20040554, Illumina Inc.) were used. The library preparation was performed according to the manufacturers' protocol (cat # 1000000124514). DNA libraries were prepared using the Twist Library Preparation EF Kit 2.0 and Twist Comprehensive Exome probe panel (cat # 104207, # 103698 and # 101308/09/10/11, Twist Bioscience) with unique dual indexes. Library preparation was performed according to the manufacturer's instructions (DOC-001239). Sequencing was carried out in the National Genomics Infrastructure (NGI) at the Science for Life Laboratory (SciLifeLab) Uppsala Sweden, using the NovaSeq X Plus system, with 150 bp paired-end read length, 25B flow cell and XLEAP-SBS sequencing chemistry. After initial quality control assessment, baseline samples with concentration at the time of extraction of less than 20 ng/μL were considered as low quantity samples and were excluded from downstream analyses.

**Whole exome sequencing preprocessing.** Whole-exome sequencing (WES) data were processed using the nf-core/sarek pipeline (v2.7)[47] employing default settings except where otherwise specified below. Paired-end FASTQ files, from multiple sequencing lanes per sample, were merged prior to analysis. Reads were aligned to the GRCh38 reference genome using BWA-MEM2 (v2.0)[48]. Post-alignment processing included duplicate marking, base quality score recalibration, and generation of recalibrated BAM files using GATK (v4.1.7.0)[49]. The mean target coverages of WES for tumor and germline DNA samples were 325x and 180x, respectively. Extensive quality control metrics, including mapping rate, duplication rate, and on-target percentage, were generated to assess sequencing and variant-calling performance.

**Somatic mutation calling.** Somatic variant calling, including single nucleotide variants (SNVs) and insertions/deletions (indels), was performed in matched tumor-normal mode using GATK4 Mutect2 (v2.2)[50], as implemented in the nf-core/sarek pipeline (v2.7). To mitigate common technical artifacts, a panel of normals (PoN) was created by running Mutect2 in tumor-only mode across all normal samples ($n = 170$), followed by merging the resulting VCF files using GATK's CreateSomaticPanelOfNormals. Mutect2 was then run on each tumor–normal sample pair using this PoN for site-level variant filtration. Variants were filtered using FilterMutectCalls with default parameters. Mutations that were present at a variant allele fraction (VAF) of less than 5%, had coverage of less than 25x in both tumor and normal samples, and were present in the gnomAD repository with a population allele frequency greater than or equal to 1% were removed. Somatic variants were annotated for potential functional effects using Ensembl Variant Effect Predictor (v113)[51] and snpEff (v4.3.1)[52]. Annotated VCF files were converted into Mutation Annotation Format (MAF) using vcf2maf (v1.6.22)[53]. Downstream analyses were restricted to non-synonymous mutations. Tumor mutational burden (TMB) was computed as the sum of all the above filtered mutations per tumor divided by the total exonic target size (36.8 Mb). APOBEC enrichment scores were estimated based on the method described by Roberts et al.[54] using the maftools R package (v2.22.0)[55].

**Copy number alteration analysis.** Copy number alterations (CNAs) were inferred using the PureCN R package (v1.22.2)[56]. A mapping bias reference was generated based on the WES bait coordinates, excluding low mappability regions. Coverage normalization was performed using a curated pool of normal blood-derived BAM files. Germline SNPs were incorporated by re-calling variants with Mutect2 in tumor-normal mode, including PoN sites, to improve allele-specific copy number estimation. The analysis workflow included Tangent normalization and segmentation steps to estimate $\log_2$ copy number ratios and to determine major and minor allele copy numbers. Significant amplifications and deletions were then identified using the Genomic Identification of Significant Targets in Cancer (GISTIC2.0) algorithm[57], applying thresholds of +0.1 for amplifications and −0.1 for deletions, with a confidence level of 0.99. Genomic events with q-values below 0.25 were considered significant at the focal or arm level. Each gene was assigned a discrete copy number state from −2 to +2, corresponding to deep deletion (−2), loss (−1), neutral (0), gain (+1), and amplification (+2). These GISTIC2.0-derived copy number states were used in downstream integrative analyses.

**RNA sequencing preprocessing and gene abundance estimation.** RNA sequencing (RNA-seq) data were processed using the nf-core/rnaseq pipeline (v3.3)[58] employing default settings except where otherwise specified below. Paired-end FASTQ files from multiple sequencing lanes per sample were merged prior to analysis. Reads were aligned to the GRch38 reference genome using the STAR aligner (v2.6.1 d)[59]. Samples exhibiting fewer than 75% of uniquely mapped reads were excluded to ensure data quality and reliability. Additional exclusions removed any libraries from patients not included in the intention-to-treat population or lacked primary endpoint data ($ORR_{12}$). Only pretreatment biopsies (baseline samples) were retained for downstream analysis.

Transcript abundance was quantified by Salmon (v1.4.0)[60]. Gene-level expression counts were derived from the transcript-level estimates using the tximport R package (v1.34.0)[61]. Gene length normalization was performed by scaling each gene's length to its geometric mean across all samples, and raw counts were then scaled by these normalized gene lengths to adjust for length biases and ensure comparability of gene counts. Effective library sizes were estimated from the scaled count matrix using the trimmed mean of M-values (TMM) normalization method via the edgeR package (v4.4.2), and log-transformed for use as offsets for downstream differential expression analysis. For visualization and downstream exploratory analyses, gene-level expression values were calculated as Transcripts Per Million (TPM), and $\log_2$-transformed to stabilize variance. All downstream analyses were performed using the resulting $\log_2$-transformed, TMM-normalized TPM matrix ($\log_2$TMM-TPM).

Gene expression outliers were identified by calculating the median Euclidean distance of each sample to all other samples based on $\log_2$-transformed gene abundance. Samples with median distances exceeding the 75th percentile plus three times the interquartile range (IQR) of all median distances were classified as outliers and excluded from downstream analyses. Finally, the remaining high-quality samples were retained for subsequent transcriptomic analysis (n = 163).

**Differential gene expression and gene set enrichment analysis.** To identify genes with altered expression between patients based on their $ORR_{12}$ status, differential expression analysis was conducted on the raw gene counts using DESeq2 R package (v1.46.0)[62]. Genes with very low counts (total raw counts ≤ 10 across all samples) were excluded prior to analysis to improve robustness. Differential expression was assessed using the Wald test, and genes with an adjusted $p$ value threshold of 0.05 (Benjamini–Hochberg correction) and an absolute $\log_2$ fold change cutoff of 0.5 were considered differentially expressed. Genes were ranked based on enrichment scores calculated as $-\log_{10}(p \text{ value}) * \text{sign}(\log_2(\text{FoldChange}))$, integrating directionality and statistical significance for pre-ranked GSEA. Gene Set Enrichment Analysis (GSEA) was performed using the fgsea R package (v1.32.0) with a minimum gene set size of 10 and a maximum of 500, using annotated Hallmark gene sets from the Molecular Signatures Database (MSigDB, v2024.1)[63]. Resulting gene sets were filtered to retain only those with an adjusted $p$ value < 0.05. Normalized enrichment scores (NES) of these significant pathways were used to explore biological processes and signaling pathways associated with the observed gene expression patterns[63]. In addition, Reactome[64] pathway enrichment analysis was performed using the ReactomePA R package (v1.50.0) with an adjusted $p$ value < 0.05. Enrichment maps were generated with the enrichplot R package (v1.26.6), incorporating pairwise pathway similarity to cluster related terms and displaying the top 25 enriched categories.

To further investigate treatment outcomes, differential expression analysis was also performed according to RCB status. As patients had received comparable therapies, the DESeq2 model included RCB status, treatment arm, and their interaction. Genes were ranked and subjected to pre-ranked GSEA with the Hallmark MSigDB gene sets, as described above, to identify pathways associated with RCB.

**Intrinsic Subtyping and Risk of Recurrence (ROR) Estimation.** Starting from the STAR-Salmon quantified data, upper-quartile normalization and $\log_2$-transformation were applied using edgeR package in accordance with[65]. Gene-centering and intrinsic subtype/ROR calculation were then performed in a single step using the BS_ssBC() function from BreastSubtypeR (v1.2.0[66]) with the parameters s = "ER.v2" and Subtype = TRUE. This function implements the subgroup-specific centering method[67], using ER + /HER2- reference quantiles[65], and automatically outputs: (a) intrinsic subtype assignment (Basal-like, HER2-Enriched, Luminal A, Luminal B), reassigning any sample whose highest correlation is to the Normal-like centroid to its next highest correlation among the four target centroids, and (b) risk of recurrence-subtype-proliferation (ROR-P) score[68].

**Estrogen receptor (ER)-related signatures.** The ESR1 module, comprising 131 genes correlated with ESR1 expression, was calculated as the average expression of these genes[69]. The ER-related gene expression module of the 21-gene Recurrence Score, including 4 ER-related genes, was computed to capture ER signaling[70]. The SET ER/PR index, based on 18 genes, was applied to measure transcription associated with ESR1 and PGR expression[71]. In addition, a research-based, in silico implementation of the ER signaling module from the NanoString Breast Cancer 360™ (BC360) panel was performed using the publicly available gene set provided by NanoString Technologies. For all signatures, samples were dichotomized at the median into high and low expression groups.

**Immune-related signatures and immune cell deconvolution.** Immune activity was assessed using complementary approaches. The Immunophenoscore (IPS) was computed using the hacksig R package (v0.1.2)[72]. The IPS provides a quantitative measure of tumor immunogenicity by aggregating biomarker scores across four categories: MHC molecules, immunomodulators, effector cells, and suppressor cells. The Danaher et al. immune cell signature method was applied according to the original publication[73], which quantifies 14 immune cell populations by averaging the expression of predefined marker genes for each subset. A tumor-infiltrating lymphocyte (TIL) score was further derived as the mean of selected immune cell scores. Immune cell abundances were also estimated using the Microenvironment Cell Populations-counter (MCP-counter) method, implemented according to the original method to quantify eight immune and two stromal populations[74]. In addition, we performed a research-based, in silico implementation of the immune-related modules available from the

NanoString Breast Cancer 360™ (BC360) panel using the publicly available gene set. The 14-gene immunoglobulin B-cell (IGG) signature was applied to capture humoral immune activity, reflecting processes such as lymphocyte maturation, CD4$^+$ and B-cell activation and survival, germinal center differentiation, immunoglobulin production, chemotaxis, and regulation of B-, T-, and NK-cell activity[75], A 9-gene and a 12-chemokine signature were applied to capture the presence of tertiary lymphoid structures (TLS)[76,77]. Beyond these signatures, expression levels of selected immune-related single genes, including human leukocyte antigens (*HLA-A, HLA-B, HLA-C, HLA-E, HLA-F, HLA-G, HLA-H, HLA-J, HLA-K, HLA-L*) and *CD274*, were also assessed. All immune-related signatures were subsequently organized into four functional categories: antigen presentation, T cell activity, B cell activity, and BC360-based immune modules. For all signatures, deconvolution-derived cell types, and single-gene measures, samples were dichotomized into high and low groups using the cohort median as a cut-off.

**Rb loss-of-function (RBsig) scoring.** The gene expression signature of Rb loss-of-function (RBsig), incorporating 87 genes that are correlated with E2F1 and E2F2 expression, was calculated for each sample as the average expression of these genes[78]. Samples with RBsig scores greater or equal to the cohort median were classified as "RBsig-high", and those below the median as "RBsig-low".

**Integrative clusters.** Integrative cluster (IntClust) subtypes, derived from gene expression and copy number data, were computed using the iC10 package (v2.0.2)[79] for samples with both data types available. IntClusts associated with the ER-positive subtype were grouped into low- (IntClusts 3, 4, 7 and 8) and high-risk (IntClusts 1, 2, 6 and 9) subgroups based on their prognostic profiles, as described in[80].

**Development and validation of the CDKPredX metagene.** A three-module metagene predictor (CDKPredX) was developed, using a proliferation-related mitosis kinase metagene score (Pr), the ER-related module of the 21-gene Recurrence Score (Er), and an immune-related kinase metagene score (Im), as they were defined in[81]. For each module, gene-level expression values (log$_2$TMM-TPM) were averaged across all module genes to yield a single module score per sample. Samples were dichotomized into high or low activity groups for each module using module-specific thresholds: the lower quartile for Pr, and the median for Er and Im. Samples with high Pr, high Er, and low Im were defined as the CDKPredX+ subgroup and the remaining as CDKPredX-. This stratification was then correlated with clinical outcomes.

The CDKPredX predictor was subsequently tested in three external validation cohorts using the same cutoffs. The first, COR-ALLEEN, is a prospective, multicenter, randomized, parallel-group, non-comparative phase II clinical trial. The trial's primary analysis and gene expression data have been previously presented[6,7]. Only patients with RNA samples available at baseline were included in the current analysis, and no quality control steps were applied to the validation cohort data. The neoadjuvant I-SPY2 trial[27] and the adjuvant SCAN-B observational study[82] are publicly available and have been previously described in detail, including gene expression data.

**Statistics and reproducibility**
This analysis concerns the primary and secondary efficacy endpoints of PREDIX LumB, as well as safety and HRQoL analysis, and protocol-predefined biomarker assessments (PET/CT, TILs, Ki67, gene expression and genomic alterations). The sample size was determined for an explorative comparison of objective response rates after 12 weeks of neoadjuvant treatment. Assuming a 40% ORR$_{12}$ in arm A (12 courses of weekly paclitaxel), and a 20% absolute improvement in arm B, a two-sided alpha = 0.10 and 80% power required 166 patients (83 per

treatment arm). The goal was thus to randomize between 180 and 200 patients in the trial[83].

Binary outcomes were compared using Fisher's exact test. Continuous variables were compared using the Wilcoxon rank-sum test. Spearman's rank correlation coefficients (with 95% bootstrap confidence intervals) quantified associations between continuous biomarkers. Logistic regression models–reporting odds ratios (ORs) with 95% confidence intervals (CIs) and Wald *p* values–assessed predictors of objective response. Time-to-event endpoints (EFS, RFS, OS) were censored at last follow-up and displayed via Kaplan–Meier curves; the log-rank test evaluated unadjusted differences. Multivariable Cox proportional hazards models–reporting hazard ratios (HRs) with 95% CIs and Wald *p* values–examined time to failure. Adjusted survival curves for the observational SCAN-B cohort were generated using the Inverse Probability of Treatment Weighting (IPTW) method, adjusted for age, size, lymph node status, and grade. Continuous biomarkers (e.g., Ki67, DKi67, TILs, DTILs, SUVmax) entered models as continuous covariates. Associations between clinicopathologic/molecular characteristics and CDKPredX subgroups were tested with Pearson's chi-squared test. HRQoL scales were analyzed via linear regression, presenting mean differences with 95% CIs and p-values. All p-values were two-sided, and the level of significance was set to 5%. Analyses were performed using Stata v17 (StataCorp, College Station, TX, USA) and R v4.4.3.

**Reporting summary**
Further information on research design is available in the Nature Portfolio Reporting Summary linked to this article.

## Data availability
The tumor and blood whole-exome sequencing data, as well as tumor RNA-sequencing data, generated in this study have been deposited in the Swedish National Data Service (SND) under SND ID: 2025-192 and are available under controlled access due to patient privacy and ethical restrictions; access may be granted upon request and following approval of a data sharing agreement. Individual-level clinical and molecular trial data cannot be made publicly available owing to restrictions imposed by the Regional Ethical Committee in Stockholm. All group- or summary-level data underlying the figures and plots are provided in the accompanying Source Data Excel file, with relevant data organized across separate sheets. Source data are provided with this paper.

## Code availability
No custom code or novel mathematical algorithms were developed for this study. All analyses relied on established, publicly available software and standard methods, which are specified in the "Methods section".

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

## Acknowledgements

The PREDIX LumB trial and correlative analyses were supported by grants from the Swedish Cancer Society (Cancerfonden), from the Swedish Breast Cancer Association (Bröstcancerförbundet), from the research funds at Radiumhemmet, and from the Swedish Research Council (Vetenskapsrådet). The study medication (palbociclib) was provided free of cost by Pfizer, which also supported the PREDIX LumB trial with an unrestricted grant. Alexios Matikas is supported by the Swedish Cancer Society (Cancerfonden), the Swedish Research Council

(Vetenskapsrådet), the Research Funds at Radiumhemmet, the Breast Cancer Association (Bröstcancerförbundet) and the Swedish Society of Medicine (Svenska Läkaresällskapet). Ioannis Zerdes is supported by Region Stockholm. Balazs Acs is supported by the Swedish Society for Medical Research and Region Stockholm. The funding sources had no access or input to the trial data, to the analyses or the manuscript. Sequencing was performed by the SNP&SEQ Technology Platform in Uppsala. The facility is part of the National Genomics Infrastructure (NGI) Sweden and Science for Life Laboratory. The SNP&SEQ Platform is also supported by the Swedish Research Council and the Knut and Alice Wallenberg Foundation. Role of the funding source: The study medication was provided free of cost by Pfizer. The funding sources had no access or input to any of the following: study design; the collection, analysis, and interpretation of data; the writing of the report; and the decision to submit the paper for publication.

## Author contributions

Concept and design: Matikas, Brandberg, Bergh (study director), Hatschek (principal investigator), Foukakis. Enrollment to the clinical trial and acquisition of clinical data: Matikas, Bjöhle, Barnekow, Margolin, Isaksson-Friman, Edman Kessler. Pathology assessment: Tzoras, Hartman, Acs, Sun, Boyaci. Wet lab: Tzoras, Salgkamis, Zerdes, Agartz. Imaging assessment: Grybäck, Zouzos. Bioinformatics and statistical analysis: Sarafidis, Sifakis, Johansson, Wang. Development of the metagene predictor: Matikas, Tzoras, Sarafidis, Sifakis, Foukakis. Drafting of the initial manuscript: Matikas, Tzoras, Sarafidis, Sifakis, Foukakis. Critical revision of the manuscript for important intellectual content: all authors. Validation: Villacampa, Pascual, Gavila, Prat, Perou. Obtained funding: Matikas, Bergh, Hatschek, Foukakis. Administrative, technical, or material support: Matikas, Hellström, Bergh, Hatschek, Foukakis. Study supervision: Matikas, Brandberg, Bergh, Hatschek, Foukakis.

## Funding

## Competing interests

A.M.: speaker (no personal fees): Roche, Seagen; consultant (no personal or institutional fees): Roche, AstraZeneca, Veracyte; research funding paid to institution by Merck, AstraZeneca, Novartis, Veracyte; advisory board: Nordic Pharma (no personal or institutional fees). I.Z.: institutional research grants from Gilead and honoraria paid to his institution from Novartis. J.H.: Leadership and ownership interests in Stratipath AB. Speaker: Gilead. Honoraria from MSD, Pfizer, Lilly. Research funding (paid to institution): Novartis, Cepheid. G.V.: speaker's fee from Pfizer, MSD, GSK and Pierre Fabre; advisory role with AstraZeneca; consultant fees from Reveal Genomics. T.P.: speaker's fee from Pfizer, AstraZeneca, Novartis, Veracyte and Argenetics; advisory role with Novartis. J.B.: research funding to institution (Karolinska Institutet and/or Karolinska University Hospital) from Amgen, AstraZeneca, Bayer, Merck, Pfizer, Roche and Sanofi; honoraria from UpToDate paid to Asklepios Medicin HB; head of advisory board at Stratipath AB; Coronis and Asklepios Cancer Research AB hold shares of Stratipath AB; honoraria for lectures/educational conferences for postgraduates courses from AstraZeneca paid to Coronis and Asklepios Cancer Research AB, chairmanship for meetings arranged by Akademikonferens on behalf of Roche, and a lecture for Novartis, honoraria to Coronis and Asklepios Cancer Research AB for both activities. T.F.: institutional fees for consultancy to AstraZeneca, Gilead and Roche; personal fees for consultancy to Affibody, Pfizer, Novartis, Veracyte, Exact Sciences; honoraria from UpToDate; research funding to institution from Pfizer, AstraZeneca, Novartis and Veracyte. All the other authors had no potential conflicts of interest to disclose.

## Additional information

[1]Oncology-Pathology Department, Karolinska Institutet, Stockholm, Sweden. [2]Breast Center, Karolinska Comprehensive Cancer Center (KCCC), Stockholm, Sweden. [3]Department of Oncology, Södersjukhuset, Stockholm, Sweden. [4]Department of Clinical Science and Education Södersjukhuset, Karolinska Institutet, Stockholm, Sweden. [5]Breast Center, St Göran Hospital, Stockholm, Sweden. [6]Department of Medical Radiation Physics and Nuclear Medicine, Karolinska University Hospital, Stockholm, Sweden. [7]Department of Clinical Pathology and Cancer Diagnostics, Karolinska University Hospital, Stockholm, Sweden. [8]SOLTI Cancer Research Group, Barcelona, Spain. [9]Statistics Unit, Vall d'Hebron Institute of Oncology, Barcelona, Spain. [10]Translational Genomics and Targeted Therapies in Solid Tumors, August Pi i Sunyer Biomedical Research Institute (IDIBAPS), Barcelona, Spain. [11]Medical Oncology Department, Hospital Clinic of Barcelona, Barcelona, Spain. [12]Faculty of Medicine and Health Sciences, University of Barcelona, Barcelona, Spain. [13]Department of Medical Oncology, Instituto Valenciano de Oncología, Valencia, Spain. [14]Lineberger Comprehensive Cancer Center, University of North Carolina at Chapel Hill, Chapel Hill, NC, USA. [15]These authors contributed equally: Alexios Matikas, Evangelos Tzoras, Michail Sarafidis. ✉e-mail: alexios.matikas@ki.se; theodoros.foukakis@ki.se

