## [Transparent Peer Review file · Nature Communications]

Neoadjuvant palbociclib and endocrine therapy versus chemotherapy in ER+/HER2- breast cancer: a randomized phase II trial

Corresponding Author: Professor Alexios Matikas

Version 0:

Reviewer comments:

Reviewer #1

(Remarks to the Author)

This is an interesting translational analysis from a randomized trial (PREDIX LumB) that attempts to identify a gene-expression signature (CDKPredX, 31 genes) predicting differential response to frontline paclitaxel versus palbociclib+endocrine therapy in ER+/HER2- breast cancer. The concept of using gene signature to select patients more likely to benefit from a CDK4/6 inhibitor-based therapy vs chemotherapy in the neoadjuvant setting is clinically relevant and of potential importance to the field.

We suggest the following points for clarification and discussion:

- Validation cohorts (Lines 70-71, 224, 233, 236-237): The authors state that the predictive signature was independently validated in the CORALLEN trial. While CORALLEN is an appropriate external cohort, the two trials differ substantially in regimen, primary endpoints, patient population, and timing of assessments, which complicates interpretation of "validation." The same caveat applies to the SCAN-B and I-SPY2 cohorts. We suggest that the authors explicitly acknowledge and discuss these differences.
- Sequential therapy discussion (Lines 101-105): The discussion of sequential CDK4/6i and chemotherapy in pancreatic cancer is interesting. Are there similar reports or pre-clinical evidence for breast cancer? (eg. PMID: 30635336)
- Subtype-specific results (Figure 2B): In the forest plot, luminal A subtype appears to favor chemotherapy, whereas luminal B favors palbociclib plus endocrine therapy, which is the opposite of what might be expected given low proliferative activity of luminal A tumors and high proliferative activity of luminal B tumors. Is there a possible explanation or hypothesis to address these results?
- Primary endpoint interpretation (Lines 274-276): The authors report no difference between chemotherapy and CDK4/6i plus endocrine therapy for the primary endpoint of ORR12 – Was the study statistically powered to detect small differences? Based on pre-specified effect size, alpha, and sample size calculation, please clarify whether this constitutes a negative trial per protocol, or whether the analysis may be underpowered. Is the ORR12 the right primary endpoint in this situation? 12 weeks of neoadjuvant ET may not be sufficient to see changes in tumor activity, and at same time, 12 weeks of weekly paclitaxel might not represent the ideal neoadjuvant chemotherapy regimen. Authors can add this in the study caveats.
- The use of neoadjuvant ET is well established for post-menopausal women and is usually performed without the use of CDK4/6i. They acknowledge this in the discussion (338-340). Have authors considered validation of CDKPredX in ET alone and how that differentiate outcomes among the two treatment modalities? This can really help highlighting the utility of CDKPredX, enabling the identification of patients who can benefit from the addition of CDK4/6 to ET alone.
- Comparison with Oncotype DX: Since Oncotype DX is the most widely use genomic assay in ER+/HER- disease, and both Oncotype and CDKPredX contain proliferation/ER-related signals, it would be informative to explore the correlation between the two signatures. If feasible, trial outcomes (chemotherapy vs. CDK4/6i+ET response) could be stratified by both Oncotype score categories (e.g., <26 vs. ≥26) and CDKPredX, which would enhance clinical interpretability.

- The use of neoadjuvant ET is not well established for pre-menopausal women. Based on the most recent discrepancy of onco-type predictive value for pre-menopausal women in the TailorX and RXPonder trials, how does CDKPredX perform in the two populations of pre- and post-menopausal women. PREDIX LumB have included both populations and interpretation may be complicated due to small sample size.
- The authors mentioned at the end of discussion that a prospective trial is undergoing to test CDKPredX as a single sample predictor. Is there a NCT number available for the trial? In what setting/population trial will run?

Reviewer #2

(Remarks to the Author)

Dear Dr. Matikas et al,

I read the manuscript titled "Randomized trial of neoadjuvant palbociclib and endocrine therapy versus chemotherapy in ER+/HER2- breast cancer (PREDIX LumB): Primary analysis and development of the CDKPredX metagene predictor" with great interest. The manuscript reports on the results of a randomized clinical trial in early stage hormone receptor positive and Her2 negative (ER+/Her2-) breast cancer with a particular emphasis on the development of a novel predictive signature to identify patients that would respond to a CDK4/6 inhibitor + endocrine therapy. Overall, it is a well written manuscript that requires minimal editing.

Key strengths of the submitted manuscript include:

1. Innovative hypothesis. The authors investigated a preoperative sequential strategy of endocrine therapy (ET) + palbociclib preceding or following chemotherapy. The trial was designed for superiority Arm B (ET + palbociclib preceding paclitaxel). Although this preoperative strategy does not have the prospect of broader clinical adoption/application, it was a relatively "clean" experiment to deduce their novel predictive signature.
2. Well conducted clinical trial with long follow up that captured at least early recurrences.
3. In-depth correlative studies.
4. Development of a signature of response to ET + CDK4/6 inhibitors challenging the paradigm that all HR+/Her2- breast cancers respond to CDK4/6 inhibitors propelled by the preconception that RB1 mutations are rare in breast cancer and RB1 mutations constitute the only mechanism of resistance to CDK4/6 inhibitors. I note the limited potential clinical application of the predictor:
 - A. neoadjuvant CDK4/6 inhibitors do not constitute standard of care. The authors appropriately discuss the potential application of this predictor to conduct such biomarker informed studies.
 - B. the predictor cannot be leveraged to identify luminal B early stage tumors that can forego chemotherapy, which constitutes a pressing and enduring question in the clinic. In other words, could patients with CDKPredX+ tumors forego chemotherapy and pursue treatment with ER + CDK4/6 inhibitors alone?

The manuscript has several limitations:

1. The protocol therapy called for adjuvant epirubicin + cyclophosphamide x 3 cycles in both arms. The authors do not report how many patients in each arm received adjuvant chemotherapy. Imbalances in adjuvant therapy may bias at least short term outcomes. The manuscript (which otherwise elaborately reports on the results of a prospective clinical trial) can benefit by adding the schema of the trial as a main figure.
2. The study used overall response rate at 12 weeks (ORR12) as its primary endpoint. Assessing ORR, especially in the 20-24% of patients with lobular carcinoma can be extremely problematic. I acknowledge that this limitation plagues all studies conducted in this setting. A subgroup analysis of ORR, PET response, and RCB in the patients with invasive lobular carcinoma would be of great interest.
3. It should be reported in the manuscript how temporally close to the 12 week biopsy and especially definitive surgery patients were allowed to take palbociclib. We know from the NeoPalANA trial that Ki67 rebounds upon discontinuation of the CDK4/6 inhibitor. Potential Ki67 rebound may skew CCA assessments.
4. How do the authors reconcile that CDKPredX+ tumors frequently harbor a high retinoblastoma loss-of-function signature? Is it because RBSig is not identical to RB1 loss? What is the association or overlap of RBSig-high and RB1 loss?
5. The CDKPredX signature is a synthesis of preexistent modules that capture features that individually were associated with differential response. The investigators did not incorporate any differential weights for the individual modules of the predictor and I am wondering whether such a strategy would enhance the discriminatory capacity of the predictor.
6. Note that the development of a predictor of response to CDK4/6 inhibitors is not new (<https://pubmed.ncbi.nlm.nih.gov/28566333/>) and prior efforts have been undertaken to identify patients who will respond to CDK4/6 inhibitors in the metastatic setting, albeit these investigations focused on individual genes of the cell cycle circuitry rather than signatures.
7. Editorial note: lines 323 – 324: "...settings where CDK4/6i have an established role, albeit concerning all-comers thus leading to both unnecessary toxicity and high costs for patients unlikely to respond." Needs to be rephrased.

Reviewer #3

(Remarks to the Author)

Reviewer #4

(Remarks to the Author)

Overall, this is an important paper and generally well-written. However, the authors should use more careful language around statistical significance, and several points require elaboration and explanation.

1. P4, L63-64: The analysis of the primary outcome does not indicate “no difference” in ORR₁₂. The two proportions, 59% vs. 45%, are meaningfully different, and the p-value (0.06) is borderline significant. It is not reasonable to say these results indicate “no difference” in the two arms on the primary outcome.
 - a. This issue is repeated several times, where “no difference” is claimed despite scientifically meaningful differences, e.g., L135-136, L234, L239.
2. P4, L70-71: The ORs in PREDIX LumB were reported – the same should be done for CORALLEEN to gauge similarity and thus the interpretation of the validation.
3. P7, L125: P-values should not be provided for Table 1 as they are not meaningful.
4. P9, L163-164: The p-value for digital TILs was < 0.05, which contradicts the statement that all interaction p-values were at least 0.1.
5. P9, L166-168: Here (and elsewhere) care is needed to note that we cannot talk about associations with pCR rates, or ORR, as each individual can experience a pCR or a response, not a rate. For instance, in this line, “lower pCR rates” should be replaced with “lower likelihood of pCR”.
6. P9, L173-175: It is not clear the units for these odds ratios – are they per 1 unit change in the ratio (i.e., for a 100% change vs. a 0% change)?
7. P10, L202-205: What does “no associations” mean in this case? Supplementary Figure 7 indicates statistically significant associations of arm with ORR by (e.g.) immunoglobulin signature, so there are not “no associations”. Perhaps this refers to interactions, not associations?
8. P10, L198-208: In general, while the interactions are not significant, there is a pattern of indications of a larger effect for “high” values of the signatures – this is worth noting, particularly since the study was not designed to look for these associations, so they may be underpowered.
9. P11, L228-229: Does this interaction relate to the outcome of objective response?
10. P12, L232-233: The ORR₁₂ proportions should also be reported for the CDKPredX- subgroup.
11. P12, L234-235: CDKPredX+ seems to be associated with ORR₂₄ and RCB, albeit not statistically significantly.
12. P12, L237-238: “...low pCR rates...” compared to what?
13. P12, L237-241: This statement is about chemoresistance, but notes that treatment in I-SPY2 was chemotherapy and/or immunotherapy – perhaps the “or” is a typo? Also, is the adjuvant treatment setting from SCAN-B relevant?
14. P12, L241-242: CDKPredX seems to have a suggestion of being prognostic for OS.
15. P13, L260-261: There is not “no difference” (e.g., constipation is borderline), but I would advise against using p-values to compare a range of baseline variables.
16. P14, L291-293: The claim “not predictive” does not seem correct as some interactions in Table 2B (and perhaps other places) were borderline significant.
17. P19, L395-396: How many people switched treatment prematurely?
18. P33, L726-735: It should be made clear that the thresholds used to create CDKPredX (that is, the medians and quartiles) were fixed before validation.
19. P33, L726-735: Presumably only individuals with all the requisite samples for the three modules were included in the derivation and evaluation of CDKPredX – how many was that in PREDIX LumB?
20. P33, L741-743: Please clarify if SCAN-B concerned the adjuvant or neoadjuvant setting.
21. P34, L763-765: Please clarify that IPTW is needed because SCAN-B is an observational study and please note whether IPTW was used in any other analyses. Also please indicate which variables were used in the IPTW model.
22. Table 2A: Please explain why paclitaxel and palbociclib are listed in both Arms A and B.

23. Table 2A: What is the p-value comparing?

Version 1:

Reviewer comments:

Reviewer #1

(Remarks to the Author)

please incorporate the analyses suggested by R#1, #7 and #8, in the supplementary results.

Reviewer #2

(Remarks to the Author)

Dear Dr. Matikas et al,

I reviewed thoroughly the revised manuscript and the responses to my comments. My comments are adequately addressed and I deem that the revisions and small additions made to the manuscript render it suitable for publication. Most importantly, it is gratifying that a prospective clinical trial to establish the clinical utility of CDKPredX is under design and the development of this biomarker will not remain just an academic exercise.

Reviewer #3

(Remarks to the Author)

Reviewer #4

(Remarks to the Author)

I thank the authors for their responses and edits. I have no further comments.

Re: Revised manuscript entitled "Randomized trial of neoadjuvant palbociclib and endocrine therapy versus chemotherapy in ER+/HER2- breast cancer (PREDIX LumB): Primary analysis and development of the CDKPredX metagene predictor" Ms. Ref. No.: NCOMMS-25-68309

We thank the reviewers for their valuable comments. We have performed revisions in the manuscript according to their suggestions. Please find hereunder our responses to the reviewers' comments, with the comment appearing in regular text and our reply following in bold text. The revised manuscript has been uploaded in two forms, one clean and one where all the changes to the manuscript are marked with the tracked changes tool.

Reviewer #1 (Remarks to the Author):

1. This is an interesting translational analysis from a randomized trial (PREDIX LumB) that attempts to identify a gene-expression signature (CDKPredX, 31 genes) predicting differential response to frontline paclitaxel versus palbociclib+endocrine therapy in ER+/HER2- breast cancer. The concept of using gene signature to select patients more likely to benefit from a CDK4/6 inhibitor-based therapy vs chemotherapy in the neoadjuvant setting is clinically relevant and of potential importance to the field. We suggest the following points for clarification and discussion:

We thank the reviewer for these comments.

2. Validation cohorts (Lines 70-71, 224, 233, 236-237): The authors state that the predictive signature was independently validated in the CORALLEN trial. While CORALLEEN is an appropriate external cohort, the two trials differ substantially in regimen, primary endpoints, patient population, and timing of assessments, which complicates interpretation of "validation." The same caveat applies to the SCAN-B and I-SPY2 cohorts. We suggest that the authors explicitly acknowledge and discuss these differences.

We agree with the reviewer that these differences comprise a study limitation, which we now acknowledge in the revised discussion (lines 350-353): *"Finally, although we took appropriate steps to externally validate our findings, we acknowledge that differences between the study cohorts in terms of chemotherapy regimens, administered CDK4/6 inhibitor in CORALLEEN, study endpoints and methods of assessment of response may have influenced our results."* **The rarity of transcriptomic datasets from randomized comparisons between chemotherapy and CDK4/6i + endocrine therapy however meant that we needed to make pragmatic decisions when we validated the gene signature.**

3. Sequential therapy discussion (Lines 101-105): The discussion of sequential CDK4/6i and chemotherapy in pancreatic cancer is interesting. Are there similar reports or pre-clinical evidence for breast cancer? (eg. PMID: 30635336)

We thank the reviewer for providing us with this highly relevant reference, which we now include to the revised manuscript. Our initial and now our updated literature search has not identified any further breast cancer-specific studies.

4. Subtype-specific results (Figure 2B): In the forest plot, luminal A subtype appears to favor chemotherapy, whereas luminal B favors palbociclib plus endocrine therapy, which is the opposite of what might be expected given low proliferative activity of luminal A tumors and high proliferative activity of luminal B tumors. Is there a possible explanation or hypothesis to address these results?

We thank the reviewer for this comment. Firstly, we cannot exclude the possibility that this is a chance finding, considering the wide confidence intervals and our inability to validate this analysis in CORALLEEN where all enrolled patients per study inclusion criteria had Luminal B tumors. In addition, the smaller NEOPAL trial enrolled patients with mostly Luminal B tumors (88%), precluding any meaningful comparison of efficacy between CDK4/6i+endocrine treatment versus chemotherapy per subtype, although the descriptive analysis doesn't point towards any differential efficacy (PMID: 30307466). Finally, CDK4/6i have proven efficacy against Luminal B disease both at the metastatic (PMID: 37939142) and adjuvant settings (increased absolute benefit from abemaciclib with higher Ki67 in monarchE, although no predictive value of Ki67 was shown), while taxane monotherapy preoperatively seems to retain its activity against Luminal A tumors (PMID: 37833133).

5. Primary endpoint interpretation (Lines 274-276): The authors report no difference between chemotherapy and CDK4/6i plus endocrine therapy for the primary endpoint of ORR12 – Was the study statistically powered to detect small differences? Based on pre-specified effect size, alpha, and sample size calculation, please clarify whether this constitutes a negative trial per protocol, or whether the analysis may be underpowered. Is the ORR12 the right primary endpoint in this situation? 12 weeks of neoadjuvant ET may not be sufficient to see changes in tumor activity, and at same time, 12 weeks of weekly paclitaxel might not represent the ideal neoadjuvant chemotherapy regimen. Authors can add this in the study caveats.

We agree with the reviewer that these are significant limitations, which we acknowledge in the discussion (lines 334-342 of the revised manuscript): *“The relatively small sample size of this randomized trial was in line with prior trials studying neoadjuvant CDK4/6i, however it may have been insufficient to detect small differences between administered treatments or associations of biomarkers with outcomes, whereas the trial was not powered for comparing the two treatment sequences for time-to-failure outcomes. Other limitations that should be acknowledged are the short treatment duration of 12 weeks with palbociclib and*

endocrine therapy which may have been insufficient to exert its full antitumoral effect and the relatively short follow-up considering the long natural history of ER+/HER2-BC, although it's one of the longest reported among neoadjuvant CDK4/6i trials.”

6. The use of neoadjuvant ET is well established for post-menopausal women and is usually performed without the use of CDK4/6i. They acknowledge this in the discussion (338-340). Have authors considered validation of CDKPredX in ET alone and how that differentiate outcomes among the two treatment modalities? This can really help highlighting the utility of CDKPredX, enabling the identification of patients who can benefit from the addition of CDK4/6 to ET alone.

We thank the reviewer for this suggestion, which we have also considered. We have applied and received approval to run the signature on the transcriptomic dataset of the PALLAS trial on adjuvant palbociclib, which is expected during the latter part of 2026. We should however underscore the fact that CDKPredX was designed to reveal chemoresistance while CDK4/6i + endocrine treatment activity is retained, and not to discern the relative contribution of palbociclib or endocrine therapy, a limitation we had already acknowledged in the discussion (lines 343-345): “Furthermore, due to the design of PREDIX LumB it is not possible to discern the relative contributions of palbociclib or endocrine therapy to treatment effect and biomarker associations”.

7. Comparison with Oncotype DX: Since Oncotype DX is the most widely use genomic assay in ER+/HER- disease, and both Oncotype and CDKPredX contain proliferation/ER-related signals, it would be informative to explore the correlation between the two signatures. If feasible, trial outcomes (chemotherapy vs. CDK4/6i+ET response) could be stratified by both Oncotype score categories (e.g., <26 vs. ≥26) and CDKPredX, which would enhance clinical interpretability.

We conducted the above analysis according to the reviewer’s comment. Since Oncotype DX had not been performed for any of the study participants, we instead ran the research version of RS from the *genefu* R package. There was weak but statistically significant correlation between CDKPredX and RS ($r=0.17$, $p=0.035$), with CDKPredX+ tumors having higher RS (Wilcoxon $p=0.036$), which was expected since both signatures include ER and proliferation genes. We then used both the proposed by the package risk cutoffs for RS, as well as the approach described in PMID: 37059742, to divide patients to high and low RS risk groups as the commercial Oncotype DX assay would. Both approaches yielded almost identical results. However, most patients in the cohort had RS high risk as shown in the table below, precluding any meaningful analysis due to low number of patients with RS low disease.

	CDKPredX+	CDKPredX-
RS high	38	100
RS low	4	21

8. The use of neoadjuvant ET is not well established for pre-menopausal women. Based on the most recent discrepancy of oncotype predictive value for pre-menopausal women in the TailorX and RXPonder trials, how does CDKPredX performs in the two populations of pre- and post-menopausal women. PREDIX LumB have included both populations and interpretation may be complicated due to small sample size.

We performed additional exploration per menopausal status as suggested by the reviewer, which is presented below. In summary, we saw no evidence of differential associations of CDKPredX with treatment effect in premenopausal versus postmenopausal women. Due to smaller patient groups, confidence intervals were wide but the point estimates remained unchanged. In support of our findings are also the results of the RIBOLARIS trial presented at ESMO 2025, where neoadjuvant CDK4/6i + endocrine treatment showed initial efficacy among premenopausal patients, as measured by the proportion of patients with ROR low at surgery.

9. The authors mentioned at the end of discussion that a prospective trial is undergoing to test CDKPredX as a single sample predictor. Is there a NCT number available for the trial? In what setting/population trial will run?

The trial is under design, while we await decisions on our applications for academic funding which is why we only briefly mention the study. Patients with ER+/HER2- breast cancer and an indication for neoadjuvant therapy will be randomized to either standard chemotherapy, or to CDKPredX-driven treatment selection (CDK4/6i+ET for positive and chemotherapy for negative patients). In total 230 patients will be randomized. The study will not start before funding is secured and CDKPredX is appropriately developed as a single sample predictor, which is why it's not yet registered.

Reviewer #2 (Remarks to the Author):

I read the manuscript titled "Randomized trial of neoadjuvant palbociclib and endocrine therapy versus chemotherapy in ER+/HER2- breast cancer (PREDIX LumB): Primary analysis and development of the CDKPredX metagene predictor" with great interest.

The manuscript reports on the results of a randomized clinical trial in early stage hormone receptor positive and Her2 negative (ER+/Her2-) breast cancer with a particular emphasis on the development of a novel predictive signature to identify patients that would respond to a CDK4/6 inhibitor + endocrine therapy. Overall, it is a well written manuscript that requires minimal editing.

Key strengths of the submitted manuscript include:

1. Innovative hypothesis. The authors investigated a preoperative sequential strategy of endocrine therapy (ET) + palbociclib preceding or following chemotherapy. The trial was designed for superiority Arm B (ET + palbociclib preceding paclitaxel). Although this preoperative strategy does not have the prospect of broader clinical adoption/application, it was a relatively “clean” experiment to deduce their novel predictive signature.
2. Well conducted clinical trial with long follow up that captured at least early recurrences.
3. In-depth correlative studies.
4. Development of a signature of response to ET + CDK4/6 inhibitors challenging the paradigm that all HR+/Her2- breast cancers respond to CDK4/6 inhibitors propelled by the preconception that RB1 mutations are rare in breast cancer and RB1 mutations constitute the only mechanism of resistance to CDK4/6 inhibitors. I note the limited potential clinical application of the predictor:
 - A. neoadjuvant CDK4/6 inhibitors do not constitute standard of care. The authors appropriately discuss the potential application of this predictor to conduct such biomarker informed studies.
 - B. the predictor cannot be leveraged to identify luminal B early stage tumors that can forego chemotherapy, which constitutes a pressing and enduring question in the clinic. In other words, could patients with CDKPredX+ tumors forego chemotherapy and pursue treatment with ER + CDK4/6 inhibitors alone?

We thank the reviewer for their thoughtful analysis. We agree with the criticisms under comment #4, essentially that we only demonstrate clinical validity but not clinical utility, which is why we plan a prospective randomized trial where decision on neoadjuvant chemotherapy will depend on CDKPredX (see our response to Reviewer 1, comment #9).

The manuscript has several limitations:

1. The protocol therapy called for adjuvant epirubicin + cyclophosphamide x 3 cycles in both arms. The authors do not report how many patients in each arm received adjuvant chemotherapy. Imbalances in adjuvant therapy may bias at least short term outcomes. The manuscript (which otherwise elaborately reports on the results of a prospective clinical trial) can benefit by adding the schema of the trial as a main figure.

Information on adjuvant EC has now been added to Table 1, per the reviewer’s suggestion. In short, EC administration was well-balanced (82 versus 82

patients completed three cycles, 3 versus 2 fewer than three and 3 versus 4 did not receive EC).

2. The study used overall response rate at 12 weeks (ORR12) as its primary endpoint. Assessing ORR, especially in the 20-24% of patients with lobular carcinoma can be extremely problematic. I acknowledge that this limitation plagues all studies conducted in this setting. A subgroup analysis of ORR, PET response, and RCB in the patients with invasive lobular carcinoma would be of great interest.

We agree with the reviewer that lobular cancer comprises a subgroup of interest in this study, which is why we have added all requested analyses in the new Supplementary Figure 14.

3. It should be reported in the manuscript how temporally close to the 12 week biopsy and especially definitive surgery patients were allowed to take palbociclib. We know from the NeoPalANA trial that Ki67 rebounds upon discontinuation of the CDK4/6 inhibitor. Potential Ki67 rebound may skew CCCA assessments.

We have now added this information to the methods section (lines 466-469) in accordance with the reviewer's request. Median time from last administered palbociclib dose to biopsy at 12 weeks (arm B) was four days, while median time from last administered palbociclib dose to surgery (arm A) was twelve days.

4. How do the authors reconcile that CDKPredX+ tumors frequently harbor a high retinoblastoma loss-of-function signature? Is it because RBsig is not identical to RB1 loss? What is the association or overlap of RBsig-high and RB1 loss?

We thank the reviewer for this highly relevant comment. The RBsig predicts E2F activity in this largely RB1 wild type cohort (only 3% had RB1 loss). This information has now been added to the main text (lines 221-223 of the revised manuscript): “In our cohort, expression of RBsig was strongly correlated with E2F signature expression ($R=0.97$, $p<0.001$). There were no RB1 point mutations, while deep deletions of RB1 gene were a rare event (5/166 patients, 3%) (Supplementary Figure 11)” and to Supplementary Figure 11.

5. The CDKPredX signature is a synthesis of preexistent modules that capture features that individually were associated with differential response. The investigators did not incorporate any differential weights for the individual modules of the predictor and I am wondering whether such a strategy would enhance the discriminatory capacity of the predictor.

We agree with the approach the reviewer suggests in their comment. Applying and validating different weights, potentially removing genes and developing a single sample predictor are our plans for the immediate future. We also refer to our response to Reviewer 1, comment #6.

6. Note that the development of a predictor of response to CDK4/6 inhibitors is not new (<https://pubmed.ncbi.nlm.nih.gov/28566333/>) and prior efforts have been undertaken to identify patients who will respond to CDK4/6 inhibitors in the metastatic setting, albeit these investigations focused on individual genes of the cell cycle circuitry rather than signatures.

We agree with the reviewer and we have now clarified our claim in the discussion that “CDKPredX is, to our knowledge, the first transcriptomic signature to predict resistance to chemotherapy while the effect of CDK4/6i and endocrine therapy is retained or even enriched”, replacing thus the previously stated more generic term “biomarker”.

7. Editorial note: lines 323 – 324: “...settings where CDK4/6i have an established role, albeit concerning all-comers thus leading to both unnecessary toxicity and high costs for patients unlikely to respond.” Needs to be rephrased.

We have now rewritten the phrase to improve clarity as such (lines 325-329 of the revised manuscript): “Further studies will shed light on the clinical validity and utility of CDKPredX at the adjuvant and metastatic settings where CDK4/6i have an established role, albeit concerning all-comers without biomarker-driven patient selection. This lack of precision may lead to both unnecessary toxicity and high costs for patients unlikely to respond.”

Reviewer #3 (Remarks to the Author):

We thank the reviewer for their contribution and wish them all the best with their future career.

Reviewer #4 (Remarks to the Author)

Overall, this is an important paper and generally well-written. However, the authors should use more careful language around statistical significance, and several points require elaboration and explanation.

1. P4, L63-64: The analysis of the primary outcome does not indicate “no difference” in ORR_12. The two proportions, 59% vs. 45%, are meaningfully different, and the p-value (0.06) is borderline significant. It is not reasonable to say these results indicate “no difference” in the two arms on the primary outcome. This issue is repeated several times, where “no difference” is claimed despite scientifically meaningful differences, e.g., L135-136, L234, L239.

We agree with the reviewer that the phrasing is unclear, since we cannot exclude with limited sample size and power the existence of a true difference in efficacy between the two arms. For clarity, we now specify in all instances that no statistically significant differences between the comparator groups were observed.

2. P4, L70-71: The ORs in PREDIX LumB were reported – the same should be done for CORALLEEN to gauge similarity and thus the interpretation of the validation.

Since we had already surpassed the word limit in the abstract by more than 10%, we have now opted to remove all odds ratios from the abstract and instead add the CORALLEEN odds ratios to the main text in the results section (besides Figure 3B).

3. P7, L125: P-values should not be provided for Table 1 as they are not meaningful.

Removed, according to the reviewer's request.

4. P9, L163-164: The p-value for digital TILs was < 0.05 , which contradicts the statement that all interaction p-values were at least 0.1.

We thank the reviewer for this comment, however we did not claim otherwise – in the original phrase we only commented on pathologist-read TILs. We have now added a phrase on digital TILs for clarity.

5. P9, L166-168: Here (and elsewhere) care is needed to note that we cannot talk about associations with pCR rates, or ORR, as each individual can experience a pCR or a response, not a rate. For instance, in this line, “lower pCR rates” should be replaced with “lower likelihood of pCR”.

Corrected throughout the manuscript, according to the reviewer's request.

6. P9, L173-175: It is not clear the units for these odds ratios – are they per 1 unit change in the ratio (i.e., for a 100% change vs. a 0% change)?

Exactly, they represent per 1 unit change of the ratio, which can be >1 in case of increased SUVmax at 12 weeks compared to baseline.

7. P10, L202-205: What does “no associations” mean in this case? Supplementary Figure 7 indicates statistically significant associations of arm with ORR by (e.g.) immunoglobulin signature, so there are not “no associations”. Perhaps this refers to interactions, not associations?

In supplementary figure 7 all confidence intervals cross 1 and all p values including tests for interaction are >0.05 . The immunoglobulin signature comes closest, with high signature correlating with response to paclitaxel (OR=0.45, 95% CI 0.18-1.09, $p=0.081$). Although we agree with the reviewer's first comment on associations and statistical significance, at the preset level of significance we could not claim statistically significant associations.

8. P10, L198-208: In general, while the interactions are not significant, there is a pattern of indications of a larger effect for “high” values of the signatures – this is worth noting, particularly since the study was not designed to look for these associations, so they may be underpowered.

We agree with the reviewer’s observation on these consistent trends, especially when it comes to immune-related and endocrine-related signatures. This is exactly the reason behind the inclusion of specific gene modules to CDKPredX during its development.

9. P11, L228-229: Does this interaction relate to the outcome of objective response?

Exactly, we have now expanded the results to clarify (lines 232-235): *“This interaction was confirmed in the CORALLEEN trial (p interaction= 0.048; Figure 3B), with the corresponding OR for the CDKPreRedX+ and CDKPreRedX- groups being OR=3, 95% CI 0.28–70.88 (p =0.395) in favor of palbociclib and endocrine therapy, and OR=0.19, 95% CI 0.04–0.71 (p =0.019) in favor of chemotherapy.”*

- 10.P12, L232-233: The ORR₁₂ proportions should also be reported for the CDKPreRedX- subgroup.

Added these results as requested by the reviewer, see our response to comment #9.

- 11.P12, L234-235: CDKPreRedX+ seems to be associated with ORR₂₄ and RCB, albeit not statistically significantly.

Especially in the case of RCB, it is difficult to say due to the, expectedly, few patients that achieved RCB 0/1. ORR₂₄ was numerically lower in the CDKPreRedX+, but as the reviewer noted the difference was not statistically significant.

- 12.P12, L237-238: “...low pCR rates...” compared to what?

We now clarify that the comparison was versus CDKPreRedX- in the trial, as shown in Supplementary Figure 12.

- 13.P12, L237-241: This statement is about chemoresistance, but notes that treatment in I-SPY2 was chemotherapy and/or immunotherapy – perhaps the “or” is a typo? Also, is the adjuvant treatment setting from SCAN-B relevant?

We have changed to “chemotherapy with or without immunotherapy” since it more accurately describes the I-SPY2 study. We used the adjuvant SCAN-B cohort for three reasons: to validate the biologic correlations of CDKPreRedX, to more accurately study the prognostic value of the signature, if any, in a very large cohort, and to add a layer of validation concerning chemoresistance of CDKPreRedX+.

- 14.P12, L241-242: CDKPreRedX seems to have a suggestion of being prognostic for OS.

We agree, which is why we further investigated its prognostic value in the much larger SCAN-B cohort which unfortunately showed no such signs (Supplementary Figure 16).

15.P13, L260-261: There is not “no difference” (e.g., constipation is borderline), but I would advise against using p-values to compare a range of baseline variables.

We appreciate the reviewer’s suggestion. The reason why we present baseline p values despite the randomized nature of the trial is to confirm that missingness in the HRQoL questionnaires was not informative. We would caution against overinterpretation of the borderline p values since we did not control the familywise error rate in these analyses.

16.P14, L291-293: The claim “not predictive” does not seem correct as some interactions in Table 2B (and perhaps other places) were borderline significant.

Although we agree with the reviewer’s observation, no other finding could be reproduced in the CORALLEEN dataset except for CDKPredX, either due to lack of data (TILs, PET/CT) or negative results (gene associations).

17.P19, L395-396: How many people switched treatment prematurely?

Rates of treatment switch due to early discontinuation of paclitaxel (arm A) and palbociclib (arm B) are given under the Safety section of the results (19.1% and 5.5%, respectively).

18.P33, L726-735: It should be made clear that the thresholds used to create CDKPredX (that is, the medians and quartiles) were fixed before validation.

We confirm that the same cutoffs were applied to all cohorts. This is now explicitly stated in the updated methods.

19.P33, L726-735: Presumably only individuals with all the requisite samples for the three modules were included in the derivation and evaluation of CDKPredX – how many was that in PREDIX LumB?

In total, 163/179 patients from the PREDIX LumB ITT population had baseline transcriptomic data and are included in all CDKPredX analyses, as described in the patient flowchart in Figure 1.

20.P33, L741-743: Please clarify if SCAN-B concerned the adjuvant or neoadjuvant setting.

Added to the methods section, as requested by the reviewer.

21.P34, L763-765: Please clarify that IPTW is needed because SCAN-B is an observational study and please note whether IPTW was used in any other analyses. Also please indicate which variables were used in the IPTW model.

Added all requested information to the Statistical analysis section of Methods. In the IPTW model we adjusted for age, size, lymph node status, and grade.

22. Table 2A: Please explain why paclitaxel and palbociclib are listed in both Arms A and B.

Since all patients switched treatment per study design, all patients from both arms were exposed to both medications. Adverse events during the respective time periods are reported in the Table.

23. Table 2A: What is the p-value comparing?

Differences between the two arms in terms of cumulative adverse events.

On behalf of all the authors,

Alexios Matikas, Associate Professor of Oncology
Department of Oncology – Pathology
Karolinska Institutet and University Hospital
17164 Stockholm, Sweden
tel: +46 767823322
e-mail: alexios.matikas@ki.se

Re: Revised manuscript entitled "Randomized trial of neoadjuvant palbociclib and endocrine therapy versus chemotherapy in ER+/HER2- breast cancer (PREDIX LumB): Primary analysis and development of the CDKPredX metagene predictor" Ms. Ref. No.: NCOMMS-25-68309

We thank the reviewers for their valuable comments. We have performed revisions in the manuscript according to their suggestions. Please find hereunder our responses to the reviewers' comments, with the comment appearing in regular text and our reply following in bold text. The revised manuscript has been uploaded in two forms, one clean and one where all the changes to the manuscript are marked with the tracked changes tool.

Reviewer #1 (Remarks to the Author):

Please incorporate the analyses suggested by R#1, #7 and #8, in the supplementary results:

These analyses have now been added to the manuscript, in the new Supplementary Figures 12 and 13.

Reviewer #2 (Remarks to the Author):

Dear Dr. Matikas et al,

I reviewed thoroughly the revised manuscript and the responses to my comments. My comments are adequately addressed and I deem that the revisions and small additions made to the manuscript render it suitable for publication. Most importantly, it is gratifying that a prospective clinical trial to establish the clinical utility of CDKPredX is under design and the development of this biomarker will not remain just an academic exercise

We thank the reviewer for their comments.

Reviewer #3 (Remarks to the Author):

We thank the reviewer for their contribution and wish them all the best with their future career.

Reviewer #4 (Remarks to the Author)

I thank the authors for their responses and edits. I have no further comments.

We thank the reviewer for their comment.

On behalf of all the authors,

Alexios Matikas, Associate Professor of Oncology
Department of Oncology – Pathology
Karolinska Institutet and University Hospital
17164 Stockholm, Sweden
tel: +46 767823322
e-mail: alexios.matikas@ki.se